# Comparing Deterministic and Soft Policy Gradients for Optimizing Gaussian Mixture Actors

**Sheelabhadra Dey**     *sheelabhadra@gmail.com*
*Department of Computer Science and Engineering*
*Texas A&M University*

**Guni Sharon**     *guni@tamu.edu*
*Department of Computer Science and Engineering*
*Texas A&M University*

**Reviewed on OpenReview:** *https://openreview.net/forum?id=qS9pPu8ODt*

## Abstract

Gaussian Mixture Models (GMMs) have been recently proposed for approximating actors in actor-critic reinforcement learning algorithms. Such GMM-based actors are commonly optimized using stochastic policy gradients along with an entropy maximization objective. In contrast to previous work, we define and study deterministic policy gradients for optimizing GMM-based actors. Similar to stochastic gradient approaches, our proposed method, denoted *Gaussian Mixture Deterministic Policy Gradient* (Gamid-PG), encourages policy entropy maximization. To this end, we define the GMM entropy gradient using *Variational Approximation* of the *KL-divergence* between the GMM's component Gaussians. We compare Gamid-PG with common stochastic policy gradient methods on benchmark dense-reward MuJoCo tasks and sparse-reward Fetch tasks. We observe that Gamid-PG outperforms stochastic gradient-based methods in 3/6 MuJoCo tasks while performing similarly on the remaining 3 tasks. In the Fetch tasks, Gamid-PG outperforms single-actor deterministic gradient-based methods while performing worse than stochastic policy gradient methods. Consequently, we conclude that GMMs optimized using deterministic policy gradients (1) should be favorably considered over stochastic gradients in dense-reward continuous control tasks, and (2) improve upon single-actor deterministic gradients.

## 1 Introduction

This study presents a comparison between deterministic and stochastic policy gradients for optimizing policies represented as Gaussian Mixture Models (GMMs) in model-free deep reinforcement learning (RL). Model-free RL was successfully demonstrated in various control domains. Examples include video games (Mnih et al., 2015; Wurman et al., 2022), robotic control (Elguea-Aguinaco et al., 2023; Dey et al., 2021; 2024; Kober et al., 2013), traffic applications (Ault & Sharon, 2021; Sharon, 2021), and medical procedures (Zhou et al., 2021; Coronato et al., 2020). Specifically, actor-critic methods (Lillicrap et al., 2015; Fujimoto et al., 2018; Haarnoja et al., 2018; Schulman et al., 2017) have proven to be effective for RL in continuous control domains. These methods were shown to be most effective when coupled with high-capacity function approximators such as neural networks for approximating both the actor and the critic.

Prior work on actor approximation can be broadly divided into two categories; (1) deterministic (Silver et al., 2014), where the policy approximator maps states to actions, and (2) soft (Haarnoja et al., 2018), where the policy approximator maps states to a distribution defined over the action space. Each of these two classes has known benefits and limitations. In particular, deterministic actors are unable to represent complex policies that capture several modes of optimal behavior while soft actors commonly assume a specific parametric distribution (often oversimplified) for which a closed-form gradient, with respect to the action distribution

parameters (e.g., mean and variance of a Gaussian), is well-defined. A line of publications (Ren et al., 2021; Peng et al., 2020; Akrour et al., 2021; Baram et al., 2021) attempted to address this gap by defining a closed-form gradient for general distributions within soft-actor optimization algorithms. These methods attempt to fit the soft actor's action distribution to the critic's action-value distribution. Empirically (on benchmark MuJuCo domains), such approaches present no, or marginal, benefits over a single-Gaussian variant (Baram et al., 2021).

To overcome the limitations associated with single Gaussian actors, Baram et al. (2021) proposed using a Gaussian Mixture Model (GMM) as the policy approximator in a maximum entropy (MaxEnt) framework. GMMs are known to be universal approximators of densities (Alspach & Sorenson, 1972) and result in a general soft actor. The proposed approach was implemented and evaluated within the Soft Actor-Critic (SAC) (Haarnoja et al., 2018) algorithm. The authors did not find clear empirical evidence supporting the use of GMMs over a single Gaussian. They attribute this finding to two reasons: "First, benchmarked tasks are unimodal in nature, so a unimodal policy should do. Second, the mixture policy collapses to a mean policy in the presence of equal mixing weights." In contrast to previous work, we propose training the GMM-based soft actor using a deep deterministic policy gradient (DDPG) approach (Lillicrap et al., 2015; Fujimoto et al., 2018). Unlike soft actor-critic methods, DDPG does not attempt to capture an underlying action-value distribution but attempts to converge on a (deterministic) local optimal action. When coupled with a GMM, such deterministic policy gradient algorithms converge on a set of local optima under common convergence conditions. However, as previously reported (Baram et al., 2021), GMM-based actors might suffer from situations where "the mixture policy collapses to a mean policy". Addressing this issue, we demonstrate how maximum entropy RL principles (Aghasadeghi & Bretl, 2011; Toussaint, 2009; Rawlik et al., 2012; Haarnoja et al., 2018; Fox et al., 2016) can be incorporated into the deterministic actor's gradient using *Variational Approximation* (Hershey & Olsen, 2007) of the KL-divergence between the GMM actor and a single Gaussian. We term our proposed approach *Gaussian Mixture Deterministic Policy Gradient* (Gamid-PG).

We conclude our study with four sets of experiments: (1) we present illustrative results on a simplified domain to showcase Gamid-PG's capability to capture multimodal behavior; (2) we conduct an empirical investigation to determine the conditions under which Gamid-PG outperforms stochastic policy gradients, and vice versa; (3) we undertake a comparative evaluation by benchmarking Gamid-PG against established deterministic and stochastic gradient-based methods in challenging continuous control environments with high-dimensional action spaces; and (4) we test the sensitivity of Gamid's hyperparameters on its performance on a representative domain.

The reported results suggest that for dense-reward control tasks with continuous action spaces, the deterministic gradient approach (Gamid-PG) can be more effective than stochastic gradient approaches to optimize GMMs and should be considered by researchers and practitioners. While, for sparse-reward tasks, stochastic gradient approaches are observed to be more effective than deterministic gradient approaches. Nevertheless, Gamid-PG is shown to improve upon single-actor deterministic gradient approaches across all the benchmark domains.

## 2 Preliminaries

### 2.1 Reinforcement learning (RL)

In RL, a policy (actuation function) is optimized over an underlying *Markov decision process* (MDP) which is a tuple $\{\mathcal{S}, \ \mathcal{A}, \ \mathcal{P}, \ R, \ \gamma, \ I_0\}$. $\mathcal{S}$ is the state space; $\mathcal{A}$ is the action space; $\mathcal{P}(s, a, s')$ is the transition probability of the form $\mathcal{P} : \mathcal{S} \times \mathcal{A} \times \mathcal{S} \rightarrow [0, 1]$, representing the probability of transitioning from state $s$ to state $s'$ after taking action $a$; $R(s, a)$ is the reward function of the form $R : \mathcal{S} \times \mathcal{A} \rightarrow \mathbb{R}$, representing the immediate utility gained from being in state $s$ and taking action $a$; $\gamma$ is the discount factor, representing the factor of lost utility in future rewards; finally, $I_0$ is a distribution over the initial state.

An RL agent is assumed to follow an internal policy, $\pi$, which maps states to actions. $\pi$ is commonly defined either as *deterministic*, i.e., $\pi : \mathcal{S} \rightarrow \mathcal{A}$, or as *soft* (stochastic), i.e., mapping states to a distribution over actions. The agent's chosen action $(a_t)$ at the current state $(s_t)$ affects the environment such that a new state emerges $(s_{t+1})$ as well as some reward $(r_t)$ representing the immediate utility gained from

performing action $a_t$ at state $s_t$, given by $R(s_t, a_t)$. $\omega$ is used to denote a finite horizon trajectory of the form $\{s_0, a_0, r_0, s_1, ..., a_{T-1}, r_{T-1}, s_T\}$. The expected *return* or expected sum of discounted rewards for a given policy is denoted by $J(\pi) = \mathbb{E}_{\omega \sim \pi} \sum_t \gamma^t r_t$. In RL the observed return is used to tune a policy such that $J(\pi)$ is maximized. The policy $\arg\max_\pi[J(\pi)]$ is the optimal policy and is denoted by $\pi^*$.[1]

**Common RL frameworks** include value-based, policy-gradient, and actor-critic approaches. A value-based approach attempts to learn the expected future utility from states (*state value*) or from action-state pairs (*action value* or *q-value*). The policy returns actions that maximize the expected utility ($\pi(s) = \arg\max_a[Q(s, a)]$). A prominent example of a value-based approach is the model-free deep $Q$-learning algorithm (DQN) (Mnih et al., 2015). In the *policy-gradient* approach (Williams, 1992) a policy is defined through a parameterized differential equation, where the policy parameters are iteratively updated, following the policy gradient, towards favorable outcomes (as experienced through the reward function). Using state or action value approximations for defining favorable outcomes for policy-gradient updates is usually referred to as an *actor-critic* approach. A prominent example of a state-of-the-art actor-critic approach is deep-deterministic-policy-gradient (DDPG) (Lillicrap et al., 2015; Fujimoto et al., 2018).

## 2.2 Distributional RL

In order to capture the intrinsic uncertainty of MDPs, a line of publications proposed to extend value-based approaches to estimate the distribution over returns. These include SPL-DQN (Luo et al., 2021), distributional-policy-gradient (Singh et al., 2022; Song & Zhao, 2020), and distributed-distributional-DDPG (D4PG) (Barth-Maron et al., 2018). These approaches still converge on a deterministic policy and, thus, might fail to capture diverse modes of optimal behavior, i.e., while the critic ($Q$-network) can capture multiple modes of optimality in the $Q$-value space, the (unimodal) actor is limited in its ability to do the same.

## 2.3 Soft policy density approximation

Another line of work suggests training a policy as a distribution over actions, i.e., a soft policy. The fitted distribution is commonly parametric to allow closed-form gradient computation. For example, training mean and variance parameters of a Gaussian. Common parametric policy distributions used in the literature include Gaussian (Schulman et al., 2015; 2017), Beta (Chou et al., 2017), and Delta (Silver et al., 2014; Lillicrap et al., 2015; Fujimoto et al., 2018). While such parametric distributions are easy to train (having closed-form gradients), they provide a limited representation power. Addressing this issue, Tessler et al. (2019) proposed *Generative Actor-Critic* (GAC). It applies *Quantile-Regression* (Koenker & Hallock, 2001) over *Autoregressive-Implicit-Quantile-Networks* (Ostrovski et al., 2018) that can represent arbitrarily densities. However, the reported results for GAC are not better (asymptotic performance and sample efficiency) than those reported for *Soft Actor-Critic* (SAC) (Haarnoja et al., 2018) which uses a single Gaussian as a soft actor policy. Another approach for capturing complex distributional properties, such as skewness, kurtosis, multimodality, and covariance structure is the *Semi-Implicit Actor* (SIA) (Yue et al., 2020). This approach adopts a *semi-implicit hierarchical construction* (Yin & Zhou, 2018) for fitting highly expressive (yet not general) parametric distributions.

## 2.4 Gaussian Mixture Model (GMM)

GMMs are probability density functions where the marginal densities of $x \in \mathbb{R}^d$ under $f$ are $f(x) = \sum_{i \in N} p_i \mathcal{N}(x; \mu_i, \Sigma_i)$, for a mixture of $N$ Gaussians where $\mathcal{N}(x; \mu_i, \Sigma_i)$ is the marginal density of $x$ for a single Gaussian, $i \in N$. $p_i$ are non-negative weighting coefficients with $\sum_n p_n = 1$. GMMs are universal approximators of densities, i.e., a GMM with sufficient components can approximate any other density function to arbitrary precision (Alspach & Sorenson, 1972).

**GMMs in RL.** GMMs were previously proposed as function approximators within RL frameworks. Agostini & Celaya (2010) proposed using GMM as $Q$-function approximators. While such an approach was shown to naturally conform to distributional RL, its overall performance (asymptotic return and sample

---

[1]In some tasks $\arg\max_\pi[J(\pi)]$ is not unique. In such cases, $\pi^*$ may refer to any optimal policy.

efficiency) is outperformed by deep-neural-network-based $Q$-approximators (Mnih et al., 2015). Another line of work (Nematollahi et al., 2022) suggested training a GMM controller using samples obtained from a SAC agent. However, the GMM controller was not integrated into the SAC algorithm (e.g., as the actor or critic), but was trained using supervised learning independently from the SAC agent. Another work (Peng et al., 2020) proposed to run multiple RL agents simultaneously with a shared reply buffer while actively encouraging policy diversity between the agents. While each agent's policy can be represented with a single Gaussian, the policy combination was not optimized as a single GMM model. Later, Baram et al. (2021) proposed the use of GMMs as policy approximators in maximum entropy (MaxEnt) frameworks. The proposed approach was implemented and evaluated within the Soft Actor-Critic (SAC) (Haarnoja et al., 2018) algorithm. The authors did not find clear evidence supporting the use of GMMs over a single Gaussian. They attribute this finding to two reasons: "First, benchmarked tasks are unimodal in nature, so a unimodal policy should do. Second, that the mixture policy collapses to a mean policy in the presence of equal mixing weights." In contrast to their claims, our experimental results (Presented in Section 4) suggest that training a GMM-based actor is advantages. In contrast to Baram et al. (2021), Ren et al. (2021) reported positive results (marginally outperforming SAC and PPO with a unimodal policy) when training parameterized mixing weights. In that work, the mixing weights are trained using a *routing function* as part of a Mixture-of-Experts RL model (Jacobs et al., 1991; Peng et al., 2019; Neumann et al., 2009; Akrour et al., 2021). Building on the partial successes of GMM-based actors in soft-actor algorithms, we extend this approach (training GMM-based actors) to a deterministic policy gradient algorithm.

## 2.5 Deep Deterministic Policy Gradient (DDPG)

DDPG (Lillicrap et al., 2015) is a benchmark continuous control RL algorithm that trains both a critic, as a differentiable $Q$-function approximator, $\hat{Q} : \mathcal{S} \times \mathcal{A} \to \mathbb{R}$, and a deterministic differentiable actor, $\pi : \mathcal{S} \to \mathcal{A}$.

**DDPG Training.** Given a randomly sampled batch of transitions, each of type $< s, a, r, s' >$, (1) the **critic** is trained to minimize the L2 TD-error (Doya, 1995), i.e., minimize:$(\hat{Q}(s, a) - (r + \gamma \hat{Q}(s', \pi(s'))))^2$; and (2) the **actor**, $\pi$, is trained to maximize $\hat{Q}(s, \pi(s))$ while assuming the $\hat{Q}$ parameters constant.

**DDPG Exploration.** An advantage of off-policy algorithms such as DDPG is that they can treat the problem of exploration independently from the trained policy. As such, DDPG performs random exploration by sampling a noise value from an *Ornstein-Uhlenbeck* process (Uhlenbeck & Ornstein, 1930) to generate temporally correlated noise. The authors do not mention a particular reason for this choice (applying correlated noise). Moreover, the authors mention that other random noise processes can be used. In our experiments with DDPG, we found that comparable results are achieved when the noise is sampled using a (simpler) non-correlated Gaussian with mean zero and variance, $\Sigma \propto I$.

## 3 Gaussian Mixture Deterministic Policy Gradient

We propose a variant of the DDPG approach termed *Gaussian Mixture Deterministic Policy Gradient* (GAMID-PG), or Gamid for short, where we (1) define the actor as a mixture of $N$ Gaussians (instead of a single Gaussian in DDPG), (2) define the actor's policy through GMM sampling, and (3) include a GMM diversification objective as part of the actor's gradient. The main motivation for the proposed algorithm is to enable deterministic policy gradients for general densities. This is justified by (1) the fact that specific types of density functions e.g., a single Gaussian, are sometimes incapable of converging to an optimal policy (Tessler et al., 2019), and (2) benefits reported for training such general-density actors in soft-actor-critic optimization (with non-deterministic policy gradients) (Ren et al., 2021). Similar to DDPG, Gamid assumes MDPs with a continuous action space as the underlying environment. Our Gamid approach is detailed in Algorithm 1 available in the form of pseudocode. The hyper(meta)-parameters include (1) the number of Gaussians for the GMM policy, $N$, (2) a shared variance for all Gaussians as a scaled identity matrix, $\Sigma$, (3) a policy divergence temperature (possibly decaying), $\tau$, and, (4) the target network update rate, $\alpha$.

---

**Algorithm 1:** Gaussian Mixture Deterministic Policy Gradient (Gamid)

---

**hyperparameters:** (1) number of Gaussians, $N$; (2) shared variance, $\Sigma \propto I$; (3) policy divergence temperature, $\tau$ (possibly decaying); (4) target update rate, $\alpha$

**init** : (1) policy parameters (for $N$ Gaussians), $\theta = \bigcup_{i=0}^{N-1} \theta[i]$; (2) $Q$-function parameters, $\phi$; (3) empty replay buffer, $\mathcal{D}$; (4) initial state, $s \sim I_0$

---

**1** Set target parameters as main parameters, $\theta_{targ} \leftarrow \theta, \ \phi_{targ} \leftarrow \phi$;

**2** **while** *not converged* **do**

**3**     Sample a Gaussian (out of $N$), $n \leftarrow \texttt{SampleGaussian}(N,P)$, where $P = \bigcup_{i \in N} p_i$ is a distribution over $N$ outcomes (e.g., uniform i.e., $\forall i, \ p_i = 1/N$) ;

**4**     Sample an action, $a \leftarrow \text{Clip}\left(\mu(s; \theta[n]) + \epsilon, \ a_{Low}, \ a_{High}\right)$, where $\epsilon \sim \mathcal{N}([0, ..., 0]^\top, \Sigma)$;

**5**     Execute $a$ in the environment, observe next state $s'$, reward $r$, and done signal $d$ indicating whether $s'$ is terminal;

**6**     Store $(s, a, r, s', d)$ in the reply buffer, $\mathcal{D}$;

**7**     Advance the environment, $s \leftarrow s'$;

**8**     **if** $d$ is TRUE, **then** reset the environment, $s \sim I_0$;

**9**     Randomly sample a batch of transitions, $B = \{(s, a, r, s', d)\}$ from $\mathcal{D}$;

**10**     Compute $Q$ targets:

$$y(r, s', d) = r + \gamma(1 - d) \max_i \left[ Q_{\phi_{targ}}\left(s', \mu(s'; \theta[i])\right) \right]$$

    Update $Q$-function with one-step gradient descent using

$$\nabla_\phi \frac{1}{|B|} \sum_{(s,a,r,s',d) \in B} \left(Q_\phi(s, a) - y(r, s', d)\right)^2$$

    **for** $i \in [0, ..., N-1]$ **do**

**11**        Update mean for Gaussian $i$, i.e., $\mu(s; \theta[i])$, with one-step gradient ascent while considering other Gaussians ($j \neq i$) constant, using

$$\nabla_{\theta[i]} \frac{1}{|B|} \left[ \sum_{s \in B} Q_\phi(s, \mu(s; \theta[i])) + \tau D_{\text{KL}}(i \| N \setminus i)) \right]$$

       where

$$D_{\text{KL}}(i \| (N \setminus i)) \approx -\log \sum_{j \neq i} \exp(-\|\mu(s; \theta[i]) - \mu(s; \theta[j])\|^2)$$

**12**     **end**

**13**     Update target parameters, $\theta_{targ} \leftarrow \alpha\theta_{targ} + (1 - \alpha)\theta, \ \text{and} \ \phi_{targ} \leftarrow \alpha\phi_{targ} + (1 - \alpha)\phi$;

**14** **end**

---

At this point, one might wonder "are GMMs still considered universal density approximators when using a single shared variance?". The answer is 'Yes'. Calcaterra (2008) showed that linear combinations of Gaussians with a single variance are, indeed, a universal density approximator. However, considering a single variance commonly requires combining more Gaussian in order to reach similar approximation accuracy levels. Nonetheless, we observed (empirically) that optimal performance for Gamid is achieved when the number of Gaussians is fairly low (2–5). Moreover, setting a constant variance allows for a practical approximation of an entropy gradient term, as described later in Section 3.2.

Given the shared variance, the GMM policy is defined solely by $N$ means. These means are approximated per Gaussian, $i \in [0, \dots, N-1]$, with a differential function approximator, $\mu_i : \mathcal{S} \times \theta[i] \to \mathbb{R}$, where $\theta[i]$ are the approximator's tunable parameters. $\theta = \bigcup_i \theta[i]$ is initialized randomly. Similarly, the $Q$-function approximator's parameters, $\phi$, are also initialized randomly.

### 3.1 GMM policy sampling

Sampling an action for a given state, $s$, from the $N$ Gaussians GMM is performed in two stages. First, (Line 3), we sample one Gaussian, $n$, from a distribution defined by the GMM weighting coefficients, $P$. Next, (Line 4), we sample an action from $n$. $P$ can be set in many ways, e.g., as a uniform distribution, $\forall i, \ p_i = 1/N$. We observed (empirically) that setting the coefficients using an $\epsilon$-greedy approach (Mnih et al., 2015) is beneficial. That is, set $p_i = 1 - \epsilon + \epsilon/N$ for $\arg\max_i Q(s, \mu(s; \theta[i]))$ and $\forall j \neq i, \ p_j = \epsilon/N$. This approach, however, introduces an extra hyperparameter, $\epsilon$, which requires meta tuning. We also experimented with existing approaches for tuning $P$ to be proportional to the $Q$-values Ren et al. (2021). However, such an approach was not observed to perform better compared to the simpler $\epsilon$-greedy approach.

### 3.2 GMM training

Known techniques for training GMM approximators from samples are mostly applicable for supervised learning, i.e., when the target GMM distribution is known (Figueiredo et al., 1999) or can be sampled (Arenz et al., 2020). Since, in our case, $\pi^*$ is not known a priori and is not available for sampling, we apply a deterministic-policy-gradient approach for training the GMM (as the actor in Gamid). The proposed gradient is defined with respect to one Gaussian ($i \in [0, \dots, N-1]$) and one state, $s$. Similar to the original DDPG algorithm, since we consider $\Sigma_i$ to be constant, the gradient is defined only with respect to the mean, $\mu(s; \theta[i])$. It is based on two (possibly conflicting) optimization objectives.

1. Maximize the expected return for a policy that follows Gaussian $i$, i.e., $\max_{\theta[i]}[Q(s; \theta[i])]$.

2. Maximize the cross-entropy between Gaussian $i$ and the GMM excluding Gaussian $i$, i.e., $\max_{\theta[i]}[\int \mathcal{N}(x; \mu(s; \theta[i]), \Sigma) \log \text{GMM}_{N \setminus i}(x; \mu(s; \theta)) dx]$

Optimization objective #1 follows the original DDPG actor training procedure. Optimization objective #2 is inspired by MaxEnt frameworks, which complement the standard maximum reward objective (Objective #1) with an entropy maximization term (Aghasadeghi & Bretl, 2011; Toussaint, 2009; Rawlik et al., 2012; Haarnoja et al., 2018; Fox et al., 2016). These two optimization objectives are presented in Line 11 of Algorithm 1 where policy $i$ is updated in the direction that increases the approximated $Q$ value AND increases the Kullback-Leibler (KL) Divergence (Kullback, 1997) from a GMM distribution which includes all the other Gaussians (other than $i$). At this point, the reader might wonder "Why maximize the KL-divergence and not the cross entropy as stated above?". There are two reasons for this choice: (1) in Gamid, KL-divergence and cross entropy are equivalent with respect to the resulting gradients; (2) it allows utilization of state-of-the-art KL-divergence approximation techniques for GMMs (Hershey & Olsen, 2007).

**From cross-entropy to KL-divergence.** Cross-entropy ($H$) is similar to KL-Divergence ($D_{\text{KL}}$) with the addition of an entropy term. More specifically, the cross-entropy of a distribution $f$ relative to a distribution $g$ is defined as $H(f, g) = D_{\text{KL}}(f \| g) + H(f)$.

**Proposition 1.** *In Gamid, cross-entropy and KL-divergence result in the same gradients with respect to a single Gaussian, $i$, and the GMM excluding $i$, denoted $GMM_{N \setminus i}$.*

*Proof.* The *Shannon entropy* of a single Gaussian, $f = \mathcal{N}(\mu, \Sigma)$, is $H(f) = 0.5 \ln \det(2\pi e \Sigma)$ (Huber et al., 2008) (note that $\pi$ refers to the mathematical constant and not a policy here). Since $H(f)$ is not a function of $\mu$, we get $\nabla_\mu H(f) = 0$. As a result, for any other density $g$ (e.g., a GMM), we have $\nabla_\mu H(f, g) = \nabla_\mu D_{KL}(f \| g)$. Note that $\nabla_\Sigma H(f, g)$ is not needed for Gamid because it assumes a constant $\Sigma$. $\qquad \square$

**Approximating $D_{\text{KL}}$.** For two distributions $f$ and $g$, $D_{\text{KL}}(f \| g)$ returns the expected log probability ratio between the two distributions, $\frac{f(x)}{g(x)}$, when $x$ is sampled from $f$. Formally, $D_{\text{KL}}(f \| g) := \mathbb{E}_{x \sim f} \log \frac{f(x)}{g(x)} = \int_x f(x) \log \frac{f(x)}{g(x)} dx$. For $k$-dimensional Gaussians $f$ and $g$ the KL-divergence has a closed form expression,

$$D_{\text{KL}}(f \| g) = \frac{1}{2} \left[ \log \frac{|\Sigma_g|}{|\Sigma_f|} + Tr[\Sigma_g^{-1} \Sigma_f] - k + (\mu_f - \mu_g)^\top \Sigma_g^{-1} (\mu_f - \mu_g) \right] \tag{1}$$

However, no closed-form expression is known for two GMMs. As a result, one might wonder "how can we compute $D_{\mathrm{KL}}$ in Gamid (Line 11)?".

Assume that $f$ and $g$ are GMMs with the following marginal densities:

$$f(x) = \sum_a p_a \mathcal{N}(x; \mu_a, \Sigma_a)$$

$$g(x) = \sum_b p_b \mathcal{N}(x; \mu_b, \Sigma_b)$$

A commonly used (closed form) approximation to $D_{\mathrm{KL}}(f\|g)$ for such cases is the *Variational Approximation* (Hershey & Olsen, 2007) which is defined as follows

$$D_{var}(f\|g) = \sum_a p_a \log \frac{\sum_{a'} p_{a'} \exp(-D_{\mathrm{KL}}(f_a\|f_{a'}))}{\sum_b p_b \exp(-D_{\mathrm{KL}}(f_a\|g_b))} \tag{2}$$

In Gamid, $f$ represents a single Gaussian (out of $N$). As such, we get a simplified $D_{var}$ expression, specifically:

$$D_{var}(f\|g) = -\log \sum_b p_b \exp(-D_{\mathrm{KL}}(f\|g_b)) \tag{3}$$

In Gamid all Gaussians comprising the GMM have a shared $\Sigma$. When setting $\Sigma = I$, i.e., the identity matrix, Equation 1 is simplified to $\|\mu_f - \mu_g\|^2$, and consequently, Equation 3 can be further simplified to

$$D_{var}(f\|g) = -\log \sum_b p_b \exp(-\|\mu_f - \mu_b\|^2) \tag{4}$$

As we seek to diversify the GMM policy overall composing Gaussians, we consider uniform weights ($\forall b$, $p_b = 1/N$) when setting $D_{var}$ in Line 11 of Gamid. As this is a constant scalar value, it is simply omitted and can be viewed as a component of the temperature scalar ($\tau$).

Finally, it is important to note, that Gamid is compatible with various DDPG variants. As such, when seeking state-of-the-art performance, one should implement it within the most effective DDPG variant. Indeed, in our experimental section, we report results for a Gamid implementation extending the Twin Delayed DDPG (TD3) variant (Fujimoto et al., 2018).

### 3.3 Convergence of Gamid

When setting the number of Gaussian to one ($N = 1$), Gamid is effectively the same as DDPG and shares similar convergence guarantees. However, when considering a mixture of $N > 1$ Gaussians, the objective function per Gaussian differs from that of DDPG following the MaxEnt sub-objective. As such, DDPG and Gamid might converge on a different local optimum. This issue can easily be addressed by decaying the policy divergence temperature ($\tau$). Nonetheless, matching the convergence guarantees of DDPG provides minor value as no such guarantees are provided for the general case. Specifically, Lillicrap et al. (2015) state, "As with $Q$ learning, introducing non-linear function approximators means that convergence is no longer guaranteed. However, such approximators appear essential in order to learn and generalize on large state spaces".

## 4 Experiments

The goals of the reported experimental study are fourfold: (G1) to illustrate the benefits of a GMM-based actor over a single Gaussian policy in the context of Gamid when attempting to capture multiple modes of optimality, (G2) to compare deterministic policy gradients versus stochastic policy gradients for optimizing a GMM-based actor, (G3) to compare Gamid against contemporary RL algorithms for continuous control tasks, and (G4) to demonstrate the performance sensitivity for Gamid's hyperparameters. Full descriptions for all the domains (state and action space, reward function) are provided in Appendix A.

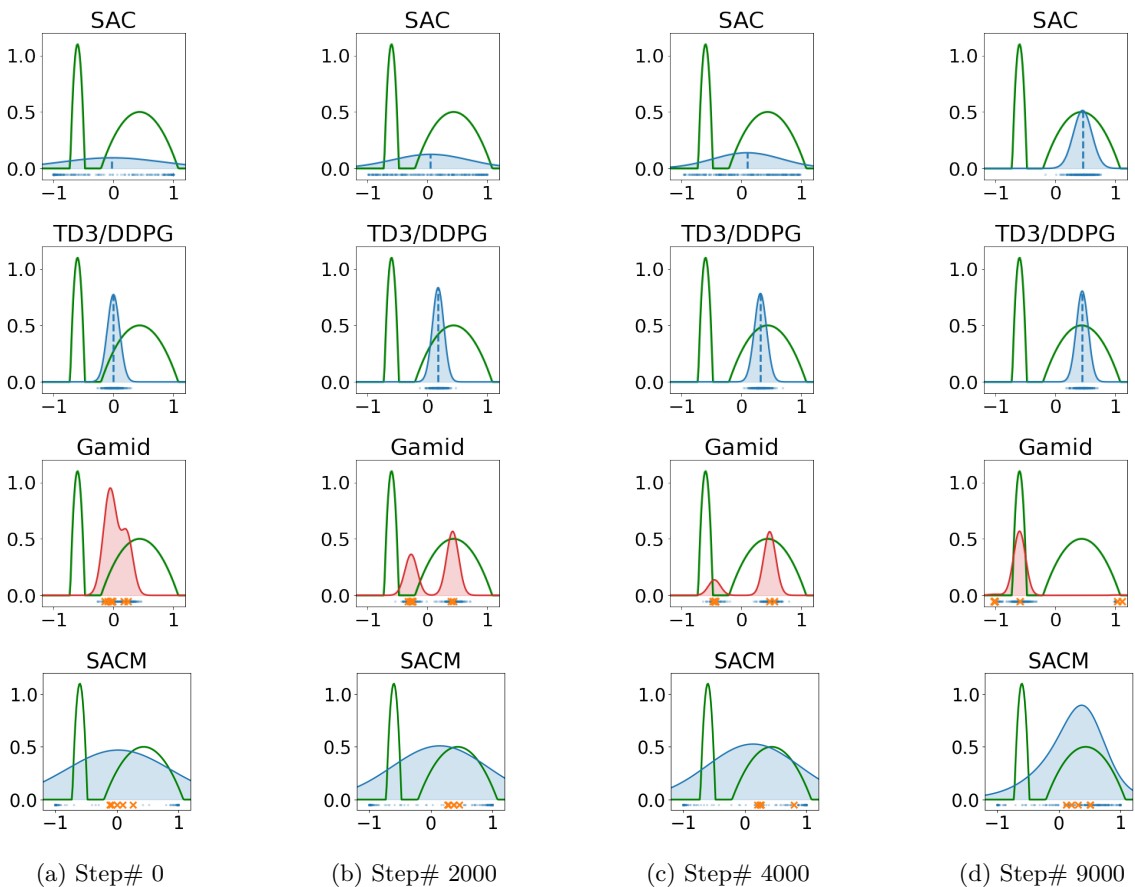

Figure 1: Policy distribution at different training steps in the 1-D continuous bandit problem with the reward function shown in green. x-axis represents the action space and y-axis the scaled PDF (probability density function). Unlike single Gaussian policy-based methods such as SAC and TD3/DDPG, and GMM policy-based SACM (shaded in blue), Gamid (shaded in red) converges on the optimal action.

## 4.1 (G1) Capturing Multiple Modes of Optimality

To illustrate the benefits of Gamid over a single Gaussian policy, as commonly used in SAC and DDPG/TD3, in terms of escaping the local optima and finding the optimal solution, we consider a toy continuous bandit problem with a 1-D action space. We adopt this problem from Huang et al. (2023). It has a multi-modal (2 modes) deterministic reward function defined over the action space in a bandit setting. The exact experimental settings follow those presented in Huang et al. (2023). For completeness, these settings are provided in Appendix A.

Figure 1 showcases the performance of Gamid against a single Gaussian policy with a parameterized mean (as in both SAC and DDPG) and standard deviation (as in SAC). The Gaussians for both SAC and DDPG are initialized with means around 0 (Figure 1a). We observe that both the single Gaussian actor variants move toward the suboptimal action (Figures 1b and 1c) and converge on it by the end of the training phase (Figure 1d). Similarly, Gamid finds the suboptimal action during the initial learning stages but, in contrast to the single Gaussian variants, it gradually spreads out and converges on the optimal action by the end of the training phase. These results suggest that a GMM-based actor can be helpful in escaping local (sub)-optimum as opposed to a single Gaussian actor. However, when examining the performance of SACM (with 5 Gaussians), we notice that it also converges on the suboptimal action, despite training a GMM actor. While SACM uses a GMM-based actor, similar to Gamid, it trains it using stochastic gradients, in contrast to Gamid. These results suggest that when training a GMM actor, deterministic gradients can be more

effective in escaping local optimum when compared to stochastic gradients. The inability of a stochastic gradient approach (SAC, SACM) to capture the optimal mode is identified in prior work [2] from which we adopted the 1-D continuous bandit environment. They provide the following explanation: "The Gaussian policy, initialized at 0 with a large standard deviation, can cover the whole solution space. However, the gradient with respect to $\mu$ is positive, which means the action probability density will be pushed towards the right, as the expected return on the right side is larger than the left side, although the left side contains a higher extreme value. As a result, the policy will move right and get stuck at the local optimum with a low chance of jumping out". We agree with the intuition provided in this explanation which highlights the limitations of stochastic gradient-based actor optimization in escaping local optimum.

### 4.1.1 GMM Actor Divergence

We analyze the diversity between the Gaussians in Gamid captured by the approximated *KL-divergence*. Specifically, we utilize Equation (4) for approximating the sum of *KL-divergence* over all Gaussians with respect to the GMM. That is, we define $\Sigma D_{var} = \sum_{i=1}^{N} D_{KL}(i \| (N \setminus i))$ (see definition from Line 11 in Algorithm 1). Figure 2 contains plots of $\Sigma D_{var}$ at each training step on three representative MuJoCo tasks (Todorov et al., 2012), 'Hopper-v3', 'HalfCheetah-v3', and 'Walker2d-v3' with increasing order of action dimensionality. We provide results for two extreme $\tau$ values (0 and 10) to highlight their effect on $\Sigma D_{var}$. We expect $\Sigma D_{var}$ to be relatively higher for higher values of $\tau$ since a higher value of $\tau$ maximizes $\Sigma D_{var}$ (Line 11 in Algorithm 1). The $\Sigma D_{var}$ curves for all the tasks in Figure 2 follow the expected trend as the curve corresponding to $\tau = 10$ is consistently higher than the one corresponding to $\tau = 0$. For comparison, we also include curves for the tuned $\tau$ values ($\tau = $ opt.) as listed in Table 5 which we observe to be between the curves corresponding to $\tau = 0$ and $\tau = 10$. These curves are closer to $\tau = 0$ since the tuned values are closer to zero at $(0.01 - 0.3)$. We further visualize the evolution of the GMM Gaussian component means in Gamid during different steps of the training stage. See Appendix D.1 for full details and results.

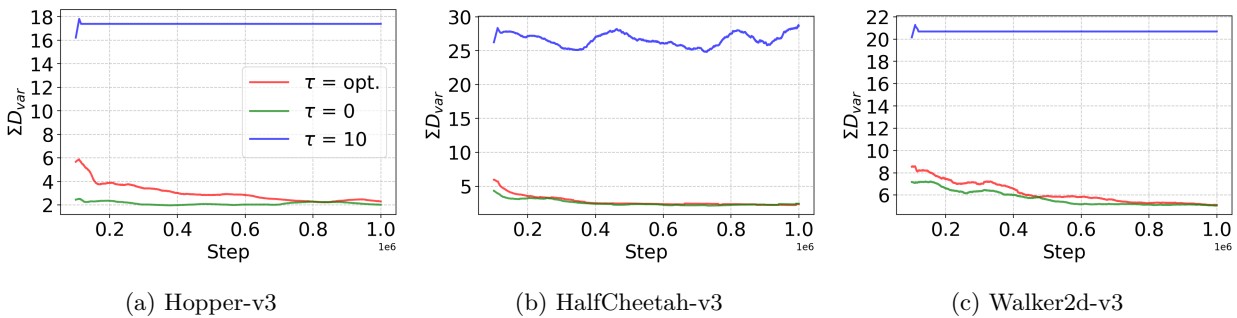

(a) Hopper-v3          (b) HalfCheetah-v3          (c) Walker2d-v3

Figure 2: Sum of the $D_{var}$ over $N$ Gaussian components ($\Sigma D_{var}$) at each training step for 3 different settings of $\tau$ on 3 representative MuJoCo tasks. Curves have been smoothed (100,000 steps moving window) for visual clarity.

## 4.2 (G2) Deterministic vs Stochastic Policy Gradients

To analyze when deterministic policy gradients can be more effective than stochastic policy we consider 2 domains. The first domain is a 2-D maze navigation grid-world environment, denoted 'MazeGrid', with a sparse reward function adopted from Huang et al. (2023). The agent starts at the center of the grid and its objective is to reach the optimal goal. The other domain is the 'HalfCheetah-v3' environment from MuJoCo (Todorov et al., 2012). Full details regarding the domains are provided in Appendix A.

In Figure 3, we observe that SACM outperforms Gamid in terms of sample efficiency by more than 1 standard deviation in 'MazeGrid'. Prior work has shown single Gaussian-based actors using MaxEnt-based policy optimization to be more effective in challenging exploration (sparse reward) tasks as compared to deterministic policies (Dawood et al., 2023; Singh et al., 2019) which might explain these trends.

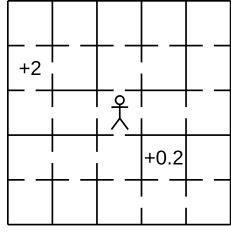 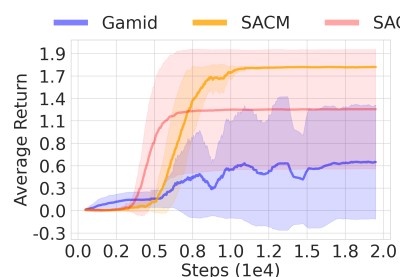

Figure 3: Training curves on MazeGrid. Solid curves present the average over five runs while the shaded region represents two standard deviations.

On the other hand, in Figure 4b we observe that Gamid outperforms SACM in terms of sample efficiency in 'HalfCheetah-v3'. Compared to stochastic policies, deterministic policies have been shown to be more effective in tasks requiring precise control (as is common in robotics domains) (Montenegro et al., 2024). We speculate that Gamid builds on these properties and uses the diversified actors to speed up exploration during the initial stages of learning. This suggests that a GMM optimized using deterministic policy gradients can be effective in dense reward robotics tasks when compared to using stochastic gradients.

### 4.3 (G3) Comparative Evaluation on Benchmark Tasks

For the comparative evaluation, we compare Gamid against common RL algorithms for continuous control tasks using benchmark MuJoCo (Todorov et al., 2012) and Fetch (Plappert et al., 2018) domains. The MuJoCo tasks utilize dense reward functions to learn locomotion tasks. The Fetch domains consist of a 7-DoF robotic arm fitted with a gripper relying on sparse reward functions to learn to solve goal-reaching tasks.

We consider the following baseline RL algorithms for continuous control tasks:

1. **Soft Actor-Critic (SAC)** (Haarnoja et al., 2018) (*Off-Policy*) – a MaxEnt actor-critic algorithm where the policy is trained to maximize a weighted combination of expected return and policy entropy.

2. **Soft Actor-Critic Mixture (SACM)** (Baram et al., 2021) (*Off-Policy*) – a soft-actor optimization approach that uses a GMM as the policy approximator. This approach was reported to not improve performance over a single Gaussian policy (SAC).

3. **Probabilistic Mixture-of-Experts SAC (PMOE)** (Ren et al., 2021) (*Off-Policy*) – a soft-actor optimization approach that uses a mixture of critics. This approach was shown to outperform other MOE approaches, specifically: MOE with gating operation (Jacobs et al., 1991), Double Option Actor-Critic (DAC) option framework (Zhang & Whiteson, 2019), the Multiplicative Compositional Policies (MCP) (Peng et al., 2019), and PMOE with Gumbel-Softmax (Maddison et al., 2016).

4. **Proximal-Policy-Optimization (PPO)** (Schulman et al., 2017) (*On-Policy*) – a *trust region policy optimization* variant (Schulman et al., 2015) using clipped gradients to restrict the policy change between policy updates.

5. **Twin Delayed DDPG (TD3)** (Fujimoto et al., 2018) (*Off-Policy*) – an extension of the original DDPG algorithm which introduces three enhancements, namely, (1) Clipped Double-$Q$ Learning, (2) Delayed Policy Updates, and (3) Target Policy Smoothing.

For Baselines 1, 4, and 5 we used the implementations provided in *Stable-baselines3* (Raffin et al., 2021). For Baseline 2, since there was no official implementation provided by the authors, we implemented it following the pseudocode provided in the paper. For Baseline 3, we used the implementation provided

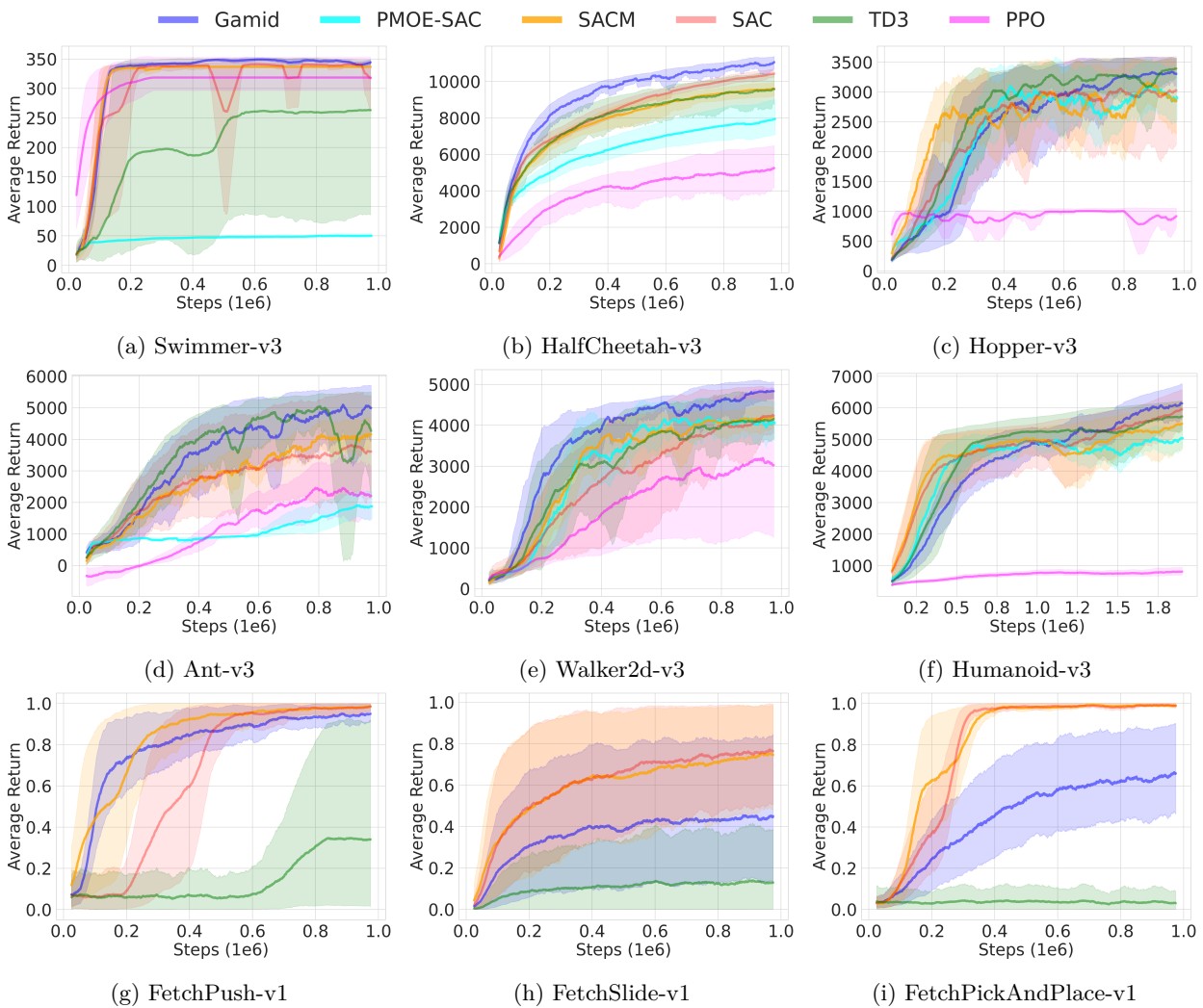

Figure 4: Training curves on continuous control benchmarks. Solid curves present the mean over five runs while the shaded region represents the tolerance interval with $\alpha = 0.05$ and $\beta = 0.7$ (Patterson et al., 2023). Curves have been smoothed (100 steps moving window) for visual clarity. Gamid (Blue curve) consistently performs on par or better compared to existing baseline methods.

by the authors. The hyperparameter values for each algorithm were set as the recommended values. For completeness, these values are provided in Appendix B. The codebase for these experiments is available at `https://github.com/Pi-Star-Lab/gamid-pg`. None of the baselines results in meaningful learning in the Fetch tasks due to sparse rewards. Consequently, we follow prior work (Ibarz et al., 2021; Raffin et al., 2021; Bajaj et al., 2023) and combine them with Hindsight Experience Replay (HER) (Andrychowicz et al., 2017). Note that PPO and PMOE are not straightforward to combine with HER and are thus omitted for these tasks. [2]

**Post-training performance.** Table 1 presents the post-training performance of Gamid and the baseline algorithms at the end of the training stage (2M training steps for 'Humanoid-v3' and 1M for the rest) on the MuJoCo and Fetch tasks. Results reported in the table and trends in Figure 5 suggest that Gamid performs on par with the baseline methods on tasks with lower degrees of freedom, as in 'Swimmer-v3' (Figure 4a), and 'Hopper-v3' (Figure 4c). It outperforms them on tasks with higher degrees of freedom (Figures 4b, 4e, 4f). We hypothesize these trends stem from increased modes of optimality present in high-

---

[2] No prior work combined PPO or PMOE with HER to the best of our knowledge.

Table 1: Mean performance and the 1-standard deviation on continuous control benchmarks. The best-performing RL algorithms have been highlighted in bold.

| | TD3 | SAC | SACM | PMOE | PPO | Gamid |
|---|---|---|---|---|---|---|
| | | | *MuJoCo (v3)* | | | |
| Swimmer | $263 \pm 139$ | $318 \pm 43$ | $336 \pm 1.4$ | $50 \pm 3.82$ | $318 \pm 22.1$ | $\mathbf{344 \pm 4.1}$ |
| HalfCheetah | $9{,}578 \pm 648$ | $10{,}427 \pm 206$ | $9{,}599 \pm 357$ | $7{,}934 \pm 1{,}072$ | $5{,}251 \pm 907$ | $\mathbf{11{,}063 \pm 300}$ |
| Hopper | $\mathbf{3{,}392 \pm 125}$ | $3{,}027 \pm 664$ | $2{,}848 \pm 677$ | $2900 \pm 461$ | $918 \pm 156$ | $3{,}301 \pm 209$ |
| Ant | $4{,}264 \pm 1{,}671$ | $3{,}613 \pm 1{,}365$ | $4{,}159 \pm 512$ | $1{,}874 \pm 491$ | $2{,}204 \pm 781$ | $\mathbf{4{,}998 \pm 828}$ |
| Walker2d | $4{,}153 \pm 429$ | $4237 \pm 429$ | $4187 \pm 245$ | $4{,}046 \pm 530$ | $3{,}023 \pm 1{,}543$ | $\mathbf{4{,}835 \pm 187}$ |
| Humanoid | $5{,}727 \pm 411$ | $5{,}949 \pm 560$ | $5{,}498 \pm 770$ | $5{,}039 \pm 394$ | $812 \pm 136$ | $\mathbf{6{,}141 \pm 629}$ |
| | | | *Fetch (v1)* | | | |
| Push | $0.34 \pm 0.43$ | $\mathbf{0.99 \pm 0.02}$ | $\mathbf{0.99 \pm 0.02}$ | — | — | $0.95 \pm 0.05$ |
| Slide | $0.13 \pm 0.18$ | $\mathbf{0.76 \pm 0.26}$ | $\mathbf{0.75 \pm 0.27}$ | — | — | $0.45 \pm 0.31$ |
| PickAndPlace | $0.03 \pm 0.04$ | $\mathbf{0.99 \pm 0.01}$ | $\mathbf{0.99 \pm 0.01}$ | — | — | $0.66 \pm 0.24$ |

dimensional control tasks and Gamid's superior ability to capture such complex behavior patterns. Results in all the Fetch tasks (Figures 4g, 4h, and 4i) show that Gamid consistently outperforms TD3 in terms of post-training performance. However, SAC and SACM outperform both Gamid and TD3. These results align with Figure 3 where we observe that stochastic gradient-based approaches are relatively more effective in sparse reward tasks than deterministic gradient-based approaches.

**Sample efficiency.** We observe that Gamid has a better sample efficiency than the baselines in 'HalfCheetah-v3' (Figure 4b) and 'Walker2d-v3' (Figure 4e). In the rest of the domains, we do not see any specific trend for Gamid as compared to the baselines. In all Fetch domains, Gamid has better sample efficiency than TD3 but has worse sample efficiency than SAC and SACM.

**Performance consistency.** In terms of the post-training performance, we observe that Gamid consistently performs at least as well as TD3 on both dense-reward (MuJoCo) and sparse-reward (Fetch) tasks. In terms of the average performance, Gamid outperforms TD3 on 8/9 tasks (see Table 1). We also report an independent two-sample $t$-test (Cressie & Whitford, 1986) with the $p$-value significance level set to 0.05 comparing Gamid and TD3. The results indicate that the advantage of Gamid over TD3 is statistically significant in 4/9 tasks ('HalfCheetah-v3', 'Walker2d-v3', 'FetchPush-v1', and 'FetchPickAndPlace-v1'). For the rest of the domains, the performance difference is not statistically significant. These results suggest that, while training with deterministic gradients, utilizing a GMM-based actor (as in Gamid) is consistently advantageous with respect to returns when compared to utilizing a single-Gaussian actor (as in TD3).

## 4.4 (G4) Sensitivity Analysis of Gamid

We examine the sensitivity of Gamid's performance with respect to its hyperparameters. The reported results exclude the hyperparameters that are shared with the original DDPG algorithm since an ablation study for those was presented in previous publications (Lillicrap et al., 2015; Fujimoto et al., 2018). The results are reported for a single domain ('Walker2d') where Gamid performs significantly better than existing approaches. Nonetheless, for completeness, ablation results for the other domains are reported in Appendix D.

**Number of Gaussians, $N$.** We start by examining the impact of varying the number of Gaussians used by Gamid (the $N$ hyperparameter). Figure 5a presents learning curves for five $N$ values: 1 (original TD3), $2, \ldots, 5$. The other hyperparameters were kept constant with values as reported in the comparative study. We observe that increasing the number of Gaussians up to $N = 3$ improves both the sample efficiency and post-training performance when compared to a single Gaussian. $N = 3$ provides the best exploration balance while the marginal benefit from adding more Gaussians diminishes and stagnates at about four. This is a

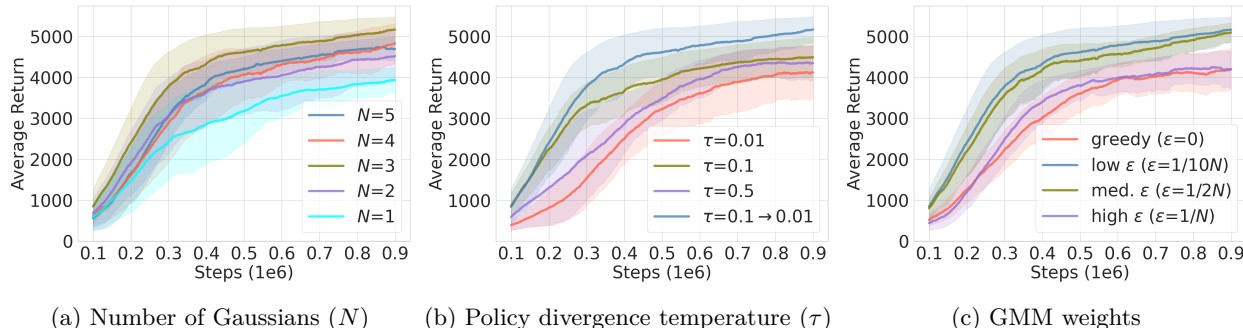

(a) Number of Gaussians ($N$)  (b) Policy divergence temperature ($\tau$)  (c) GMM weights

Figure 5: Ablation curves for the 'Walker2D' domain. These results suggest that (1) adding more Gaussians to the GMM-based actor is beneficial, (2) setting a decaying policy divergence parameter is beneficial, and (3) using an $\epsilon$-greedy approach with a low (yet not zero) $\epsilon$ value is beneficial.

reasonable result as, with sufficient Gaussians, the GMM becomes expressive enough to represent any target policy, so the addition of more Gaussians is not helpful.

**Policy divergence temperature, $\tau$.** Next, we examine the impact of varying the policy divergence temperature hyperparameter ($\tau$). Figure 5b presents learning curves for three static $\tau$ values (0.01, 0.1, 0.5) and a decaying $\tau$ version (linear decay from 0.1 to 0.01 in 30% of the total training steps). We observe that a decaying $\tau$ value performs best. This outcome (favoring entropy temperature decay) is in line with similar results reported by Schulman et al. (2017) and Haarnoja et al. (2018).

**GMM weights, $\epsilon$.** Finally, we examine the impact of varying the GMM weights setting from Line 3 (denoted $P$). Figure 5c presents learning curves for four $P$ $\epsilon$-greedy assignments: $\epsilon = 0$ (greedy), $\epsilon = 1/(10N)$ (low epsilon), $\epsilon = 1/(2N)$ (high epsilon), $\epsilon = 1/N$ (uniform). The results suggest that a low (yet not zero) epsilon performs best. This result is in line with $\epsilon$-greedy trends reported in prior work (Mnih et al., 2015).

## 5 Discussion

While prior works have compared deterministic and stochastic policy gradients they did so for single Gaussian actors. These studies found that stochastic gradients outperform deterministic gradients on common benchmark domains (Haarnoja et al., 2018). Our experimental study suggests that this trend (stochastic gradients outperform deterministic gradients) does not necessarily apply to GMM-based actors. We observe that in 5 out of 6 dense-reward MuJoCo domains, optimizing such actors with deterministic gradients (Gamid) performed better compared to stochastic gradients (SACM). Gamid also leads to improvements over (single Gaussian) SAC in 3/6 MuJoCo tasks while consistently performing similar or better than TD3 in all the MuJoCo tasks. In the 3 sparse-reward Fetch tasks, we observe that stochastic gradient-based actors (SAC, SACM) outperform deterministic gradient-based actors (GMM). In stochastic gradient-based algorithms, the entropy of the policy, derived from a learned standard deviation as opposed to a fixed standard deviation in deterministic gradient-based algorithms, is generally high during the initial learning stages in sparse-reward tasks. During this stage, the agent receives close to zero non-zero rewards that result in high policy standard deviation and hence a close-to-random exploration which is key in such scenarios. We speculate that such a property makes SAC and SACM more effective as compared to Gamid and TD3 in the Fetch tasks. Nonetheless, Gamid outperforms TD3 in all the Fetch tasks, suggesting that using stochastic gradients to train a GMM actor can be more effective than doing the same over a single Gaussian actor. We observe that GMM-based actors optimized using deterministic gradients as presented in Gamid do not adversely affect the post-training performance of TD3 in the MuJoCo tasks. These results stand in contrast with findings reported in Baram et al. (2021) that did not find any significant advantage for using GMM-based actors.

It should be noted that Gamid adds additional hyperparameters on top of DDPG/TD3 which may raise concerns about its practicality in real-world settings, given the need for required hyperparameter tuning. We emphasize that Gamid, in its current form, is an effort to spark interest within the scientific community on the effectiveness of deterministic policy gradients when coupled with GMMs. Future work will focus on developing practical variants of Gamid by fixing or automating hyperparameter values.

## 6 Summary

In this paper, we presented a comparison between stochastic and deterministic policy gradients to optimize Gaussian mixture model (GMM)-based policies. We introduced a novel approach, denoted Gamid, for training a GMM using deterministic gradients. Similar to the maximization of entropy in stochastic actors, Gamid incorporates a diversification objective that encourages the actors to spread out aiding the exploration capabilities of the policy. Empirical studies on benchmark MuJuCo tasks show that, in terms of sample efficiency and post-training performance, Gamid improves over the single Gaussian stochastic variant (SAC) in 3/6 domains. It consistently performs on par or better than the single Gaussian deterministic variant (TD3) in all the MuJoCo and Fetch tasks. These results suggest that deterministic gradient approaches can be more effective for training GMM actors as compared to stochastic gradient approaches for dense-reward control tasks. Empirical results on sparse-reward Fetch tasks show that stochastic gradient approaches are more effective than Gamid. Nevertheless, Gamid improves over the single Gaussian deterministic variant in 3/3 Fetch domains. Consequently, we hope this work will seed research on training soft actors using deterministic gradient approaches.

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

## A Domain descriptions

**1-D Continuous Bandit:** A bandit problem with a continuous 1-dimensional action space as shown in Figure 6. It has a 1-D action space and a non-convex reward landscape with 2 modes of optimality; a global and a local.

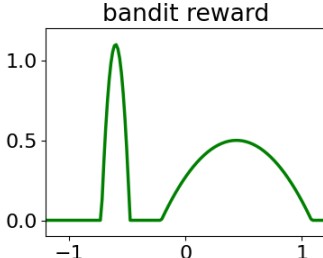

Figure 6: 1-D Continuous Bandit environment

**MazeGrid:** A 2D maze navigation task as shown in Figure 7. The maze consists of 5×5 grids. Each of them is connected with neighbors with a narrow passage. The agent starts in the center grid and can move in four directions. The action space is its position change in two directions $(\Delta x, \Delta y)$.

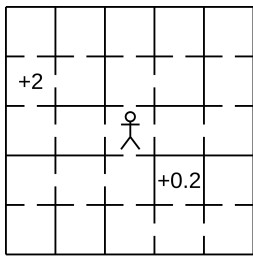

Figure 7: MazeGrid environment

**MuJoCo:** A comprehensive domain description for all the MuJoCo (v3) tasks along with installation and exact instructions can be found through: `https://gymnasium.farama.org/environments/mujoco/`. Table 2 provides a summary of the state (observation) and action spaces dimensionality along with the guiding reward function definition per task.

## B Hyperparameters

We adapted the 1-D continuous bandit environment introduced in prior work (Huang et al., 2023) and used the recommended hyperparameters for REINFORCE for setting up a lightweight version of SAC (no critic and no replay buffer). We also use a similar lightweight version of DDPG/TD3 (no critic and no

Table 2: (a) Action space degrees-of-freedom (Action dim.), (b) Observation space dimensionality (Obs. dim.), (c) reward function (Reward), per domain.

| Domain | Action dim. | Obs. dim. | Reward |
|---|---|---|---|
| Swim. | 2 | 8 | forward_reward - ctrl_cost |
| HalfCh. | 6 | 17 | forward_reward - ctrl_cost |
| Hopper. | 3 | 11 | healthy_reward + forward_reward - ctrl_cost |
| Ant. | 8 | 27 | healthy_reward + forward_reward - ctrl_cost - contact_cost |
| Walker. | 6 | 17 | healthy_reward + forward_reward - ctrl_cost |
| Human. | 17 | 376 | healthy_reward + forward_reward - ctrl_cost - contact_cost |
| Push. | 4 | 25 | sparse; block final target position reached = 0/not reached = -1 |
| Slide. | 4 | 25 | sparse; puck final target position reached = 0/not reached = -1 |
| Pick. | 4 | 25 | sparse; block final target position reached = 0/not reached = -1 |

replay buffer). For the noise in DDPG, we used a Gaussian distribution with a mean of 0 and a fixed standard deviation of 0.1, the recommended setting for MuJoCo tasks in *RL Baselines3 Zoo* library (`https://github.com/DLR-RM/rl-baselines3-zoo/tree/v1.6.2/hyperparams`).

Tables 8–9 present the default hyperparameters values used in our experiments on the MuJoCo and Fetch tasks. To allow for easy reproduction of the results, the presented hyperparam names follow the variable names in the *Stable-baselines3 (v1.6.2)* (Raffin et al., 2021) open-source codebase. We used the tuned hyperparameter values for SAC, TD3, and PPO that have been provided in *RL Baselines3 Zoo* which is built on top of *Stable-baselines3*. For SACM and PMOE we used hyperparameters recommended in (Baram et al., 2021) and (Ren et al., 2021) respectively for the MuJoCo tasks. For the Fetch tasks, we tuned the hyperparameters of SACM using Optuna (Akiba et al., 2019).

Table 3: Shared TD3 hyperparameters (MuJoCo)

| Hyperparam. | Swimmer-v3 | HalfCheetah-v3 | Hopper-v3 | Ant-v3 | Walker2d-v3 | Humanoid-v3 |
|---|---|---|---|---|---|---|
| n_timesteps | 1000000 | 1000000 | 1000000 | 1000000 | 1000000 | 2000000 |
| learning_rate | 0.001 | 0.001 | 0.0003 | 0.001 | 0.001 | 0.0003 |
| learning_starts | 10000 | 10000 | 10000 | 10000 | 10000 | 10000 |
| batch_size | 100 | 100 | 256 | 100 | 100 | 256 |
| gamma | 0.9999 | 0.99 | 0.99 | 0.99 | 0.99 | 0.99 |
| train_freq | 1 | (1, episode) | 1 | (1, episode) | (1, episode) | 1 |
| gradient_steps | 1 | $-1$ | 1 | $-1$ | $-1$ | 1 |
| noise_type | 'normal' | 'normal' | 'normal' | 'normal' | 'normal' | "normal" |
| noise_std | 0.1 | 0.1 | 0.1 | 0.1 | 0.1 | 0.1 |
| net_arch | [256, 256] | [256, 256] | [256, 256] | [256, 256] | [256, 256] | [256, 256] |
| activation_fn | nn.ReLU | nn.ReLU | nn.ReLU | nn.ReLU | nn.ReLU | nn.ReLU |

## C   Compute

All reported experiments were distributed between 2 machines; (1) a machine with 64 32-core AMD Ryzen Threadripper PRO 5975WX CPUs, each clocked at 4.3 GHz with 250 GB RAM with 2 NVIDIA GeForce RTX 3090 24 GB GPUs (2) a machine with 16 8-core Intel(R) Core(TM) i7-9800X CPUs, each clocked at 3.8 GHz with 16 GB RAM and an NVIDIA GeForce RTX 20280 Ti 12 GB GPU.

## D   Additional experiments

### D.1   Gaussian Spread

Figures 8 and 9 (Appendix D.1) show the spread of the GMM Gaussian means at different steps in the training stage. Since the action dimensionality in all of the MuJoCo environments is greater than two, we

Table 4: Shared TD3 hyperparameters (Fetch)

| Hyperparam. | FetchPush-v1 | FetchSlide-v1 | FetchPickAndPlace-v1 |
|---|---|---|---|
| n_timesteps | 1000000 | 1000000 | 1000000 |
| learning_rate | 0.001 | 0.001 | 0.001 |
| learning_starts | 1000 | 1000 | 1000 |
| batch_size | 2048 | 2048 | 1024 |
| gamma | 0.95 | 0.95 | 0.95 |
| noise_type | 'normal' | 'normal' | 'normal' |
| noise_std | 0.1 | 0.1 | 0.1 |
| net_arch | $[512, 512, 512]$ | $[512, 512, 512]$ | $[512, 512, 512]$ |
| activation_fn | nn.ReLU | nn.ReLU | nn.ReLU |
| replay_buffer_class | HerReplayBuffer | HerReplayBuffer | HerReplayBuffer |
| replay_buffer_kwargs | 'dict(online_sampling=True, goal_selection_strategy='future', n_sampled_goal=4)' | | |

Table 5: Gamid-specific hyperparameters

| Hyperparam. | Swim. | HalfCh. | Hopper. | Ant. | Walker. | Human. | Push. | Slide. | Pick. |
|---|---|---|---|---|---|---|---|---|---|
| n_actors ($N$) | 3 | 5 | 2 | 2 | 3 | 4 | 4 | 8 | 8 |
| temperature_initial ($\tau$) | 0.1 | 0.1 | 0.1 | 0.1 | 0.1 | 0.3 | 0.3 | 0.3 | 0.3 |
| temperature_final ($\tau$) | 0.1 | 0.1 | 0.01 | 0.1 | 0.1 | 0.1 | 0.1 | 0.1 | 0.1 |
| temperature_fraction | 1.0 | 1.0 | 1.0 | 1.0 | 1.0 | 0.3 | 1.0 | 1.0 | 1.0 |
| epsilon_initial ($\epsilon$) | 0.3 | 0.2 | 0.1 | 0.1 | 0.3 | 0.3 | 0.1 | 0.1 | 0.1 |
| epsilon_final ($\epsilon$) | 0.03 | 0.02 | 0.1 | 0.1 | 0.03 | 0.1 | 0.01 | 0.01 | 0.01 |
| epsilon_fraction | 0.5 | 0.5 | 1.0 | 1.0 | 1.0 | 0.3 | 1.0 | 1.0 | 1.0 |

perform a principal component analysis (PCA) (Abdi & Williams, 2010) on the Gaussian means and plot the top two PCA components. Comparing the top and middle rows of Figures 8 corresponding to $\tau = 0$ and $\tau = 10$ in 'Hopper-v3' respectively, we observe that the Gaussian means have a higher spread for $\tau = 10$. A similar observation can also be made when comparing the top and middle rows of Figures 9 corresponding to $\tau = 0$ and $\tau = 10$ in 'HalfCheetah-v3' respectively. The trend in bottom rows Figures 8 and 9 corresponding to $\tau = $ opt. is less clear which we speculate could be attributed to noisy weight updates due to mini-batch gradient descent during training.

### D.2 Sensitivity analysis for MuJoCo tasks

Following Haarnoja et al. (2018), we include the sensitivity analysis for all the MuJoCo tasks (beyond the representative, single task, results presented in the main text).

Figures 10–15 present training curves similar to those presented in the ablation study from the main text albeit on all the MuJoCo tasks. These results generally support the conclusions provided in the main text (1) adding more Gaussians to the GMM-based actor has a positive impact up to a certain threshold, beyond which, performance stagnates or even deteriorates, (2) setting a decaying policy divergence parameter is beneficial, and (3) using an $\epsilon$-greedy approach with a low (yet not zero) $\epsilon$ value is beneficial.

Table 6: SAC hyperparameters (MuJoCo)

| Hyperparam. | Swimmer-v3 | HalfCheetah-v3 | Hopper-v3 | Ant-v3 | Walker2d-v3 | Humanoid-v3 |
|---|---|---|---|---|---|---|
| n_timesteps | 1000000 | 1000000 | 1000000 | 1000000 | 1000000 | 2000000 |
| learning_rate | 0.0003 | 0.0003 | 0.0003 | 0.0003 | 0.0003 | 0.0003 |
| learning_starts | 10000 | 10000 | 10000 | 10000 | 10000 | 10000 |
| gradient_steps | 1 | 1 | 1 | 1 | 1 | 1 |
| gamma | 0.9999 | 0.99 | 0.99 | 0.99 | 0.99 | 0.99 |
| tau | 0.05 | 0.05 | 0.05 | 0.05 | 0.05 | 0.05 |
| net_arch | [256, 256] | [256, 256] | [256, 256] | [256, 256] | [256, 256] | [256, 256] |
| activation_fn | nn.ReLU | nn.ReLU | nn.ReLU | nn.ReLU | nn.ReLU | nn.ReLU |

Table 7: SACM hyperparameters (MuJoCo)

| Hyperparam. | Swimmer-v3 | HalfCheetah-v3 | Hopper-v3 | Ant-v3 | Walker2d-v3 | Humanoid-v3 |
|---|---|---|---|---|---|---|
| n_components | 3 | 3 | 3 | 3 | 3 | 3 |
| learning_rate | 0.0006 | 0.0003 | 0.0003 | 0.0003 | 0.0003 | 0.0003 |
| learning_starts | 10000 | 10000 | 10000 | 10000 | 10000 | 10000 |
| gradient_steps | 1 | $-1$ | 1 | $-1$ | $-1$ | 1 |
| tau | 0.05 | 0.05 | 0.05 | 0.05 | 0.05 | 0.05 |
| net_arch | [256, 256] | [256, 256] | [256, 256] | [256, 256] | [256, 256] | [256, 256] |
| activation_fn | nn.ReLU | nn.ReLU | nn.ReLU | nn.ReLU | nn.ReLU | nn.ReLU |

Table 8: SAC and SACM hyperparameters (Fetch)

| Hyperparam. | FetchPush-v1 | FetchSlide-v1 | FetchPickAndPlace-v1 |
|---|---|---|---|
| n_timesteps | 1000000 | 1000000 | 1000000 |
| learning_rate | 0.001 | 0.001 | 0.001 |
| learning_starts | 1000 | 1000 | 1000 |
| batch_size | 2048 | 2048 | 1024 |
| gamma | 0.95 | 0.95 | 0.95 |
| ent_coef | 'auto' | 'auto' | 'auto' |
| net_arch | [512, 512, 512] | [512, 512, 512] | [512, 512, 512] |
| activation_fn | nn.ReLU | nn.ReLU | nn.ReLU |
| n_components (SACM) | 3 | 3 | 3 |
| replay_buffer_class | HerReplayBuffer | HerReplayBuffer | HerReplayBuffer |
| replay_buffer_kwargs | 'dict(online_sampling=True, goal_selection_strategy='future', n_sampled_goal=4)' | | |

Table 9: PPO hyperparameters (MuJoCo)

| Hyperparam. | Swimmer-v3 | HalfCheetah-v3 | Hopper-v3 | Ant-v3 | Walker2d-v3 | Humanoid-v3 |
|---|---|---|---|---|---|---|
| normalize | True | True | True | True | True | True |
| gamma | 0.9999 | 0.98 | 0.999 | 0.99 | 0.99 | 0.95 |
| n_steps | 1024 | 512 | 512 | 2048 | 512 | 512 |
| batch_size | 256 | 64 | 32 | 64 | 32 | 256 |
| learning_rate | 0.0006 | $9.8e{-}5$ | $3e-4$ | 0.0001 | $5.05e-5$ | $3.56e-5$ |
| ent_coef | $4.02e4$ | $4e{-}4$ | 0.0022 | 0.0 | $5.8e-4$ | 0.0024 |
| clip_range | 0.9999 | 0.1 | 0.2 | 0.2 | 0.2 | 0.3 |
| n_epochs | 20 | 20 | 5 | 10 | 20 | 5 |
| gae_lambda | 0.98 | 0.92 | 0.99 | 0.95 | 0.95 | 0.9 |
| max_grad_norm | 0.8 | 0.8 | 0.7 | 0.5 | 1 | 2 |
| vf_coef | $4.02e^4$ | 0.58 | 0.83 | 0.5 | 0.87 | 0.43 |
| net_arch | $[256, 256]$ | $[256, 256]$ | $[256, 256]$ | $[256, 256]$ | $[256, 256]$ | $[256, 256]$ |
| activation_fn | nn.ReLU | nn.ReLU | nn.ReLU | nn.ReLU | nn.ReLU | nn.ReLU |
| log_std_init | 0.0 | $-2$ | $-2$ | 0.0 | 0.0 | $-2$ |
| ortho_init | True | False | False | True | True | False |

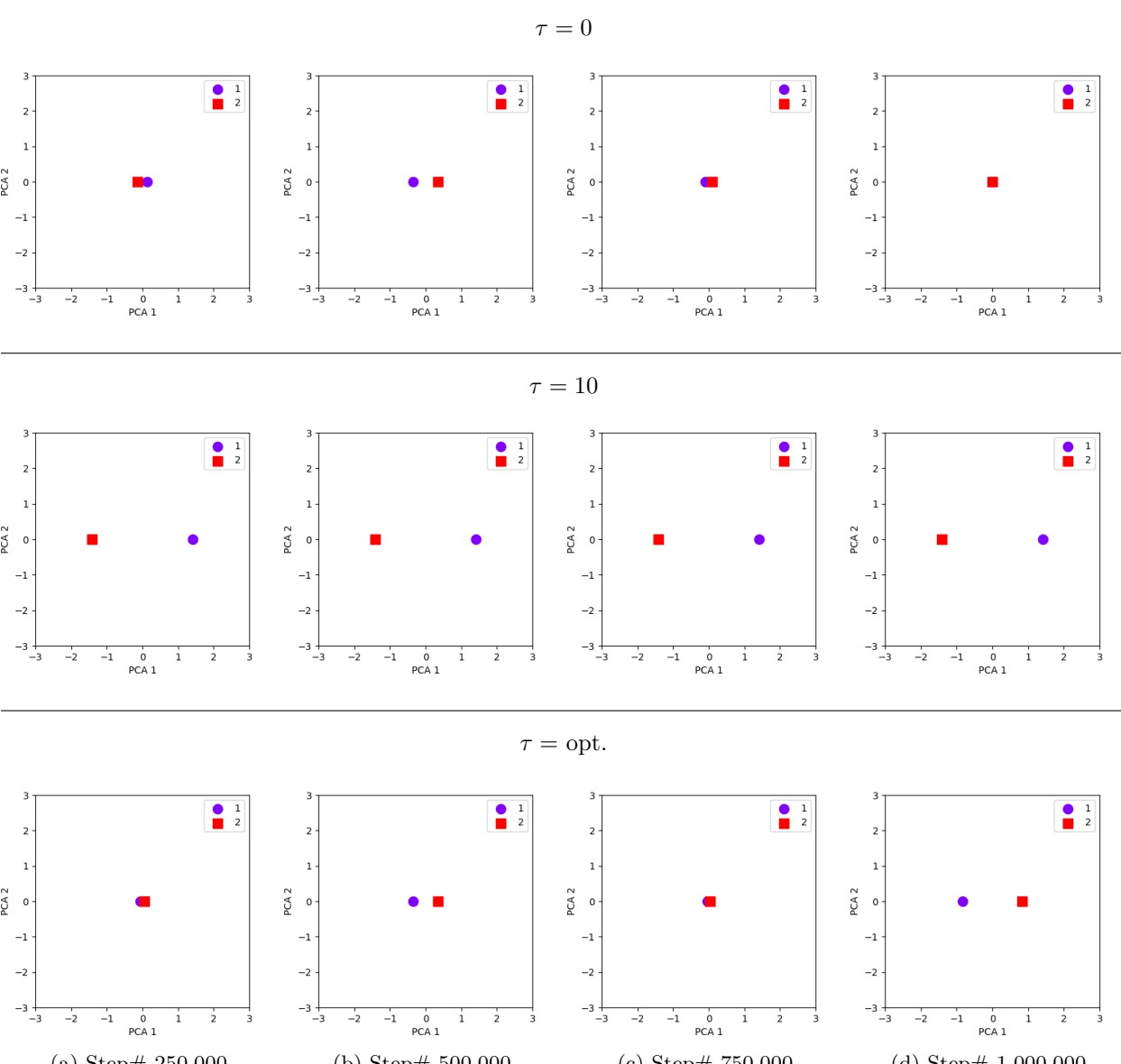

Figure 8: Principal component analysis (PCA) visualization of Gamid Gaussian means in 'Hopper-v3' with $\tau = 0$ (top row), $\tau = 10$ (middle row) and $\tau$ set to the tuned value (bottom row). The legend represents the ID of each Gaussian.

$\tau = 0$

$\tau = 10$

$\tau = \text{opt.}$

(a) Step# 250,000     (b) Step# 500,000     (c) Step# 750,000     (d) Step# 1,000,000

Figure 9: Principal component analysis (PCA) visualization of Gamid Gaussian means in 'HalfCheetah-v3' with $\tau = 0$ (top row), $\tau = 10$ (middle row) and $\tau$ set to the tuned value (bottom row). The legend represents the ID of each Gaussian.

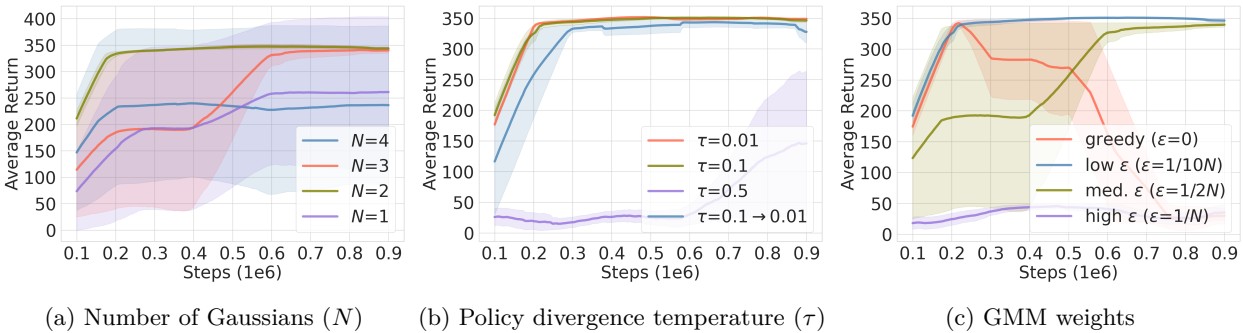

(a) Number of Gaussians ($N$)     (b) Policy divergence temperature ($\tau$)     (c) GMM weights

Figure 10: Ablation curves for the 'Swimmer-v3' domain

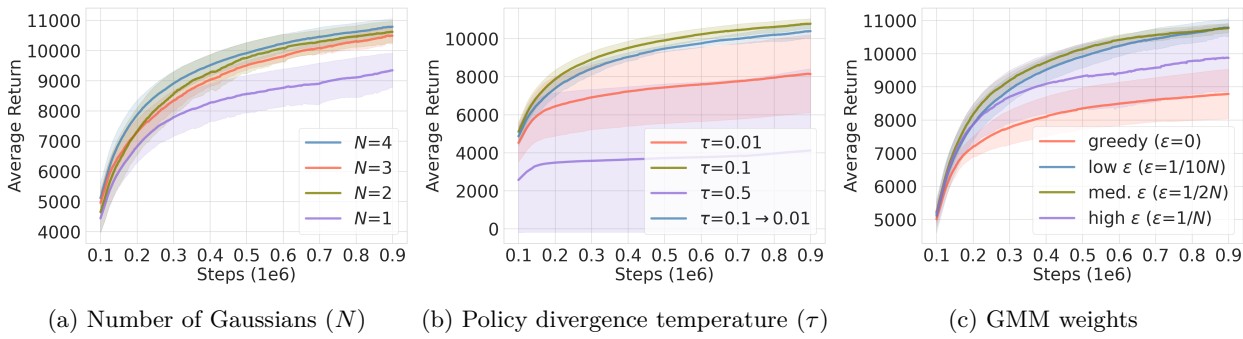

(a) Number of Gaussians ($N$)      (b) Policy divergence temperature ($\tau$)      (c) GMM weights

Figure 11: Ablation curves for the 'HalfCheetah-v3' domain

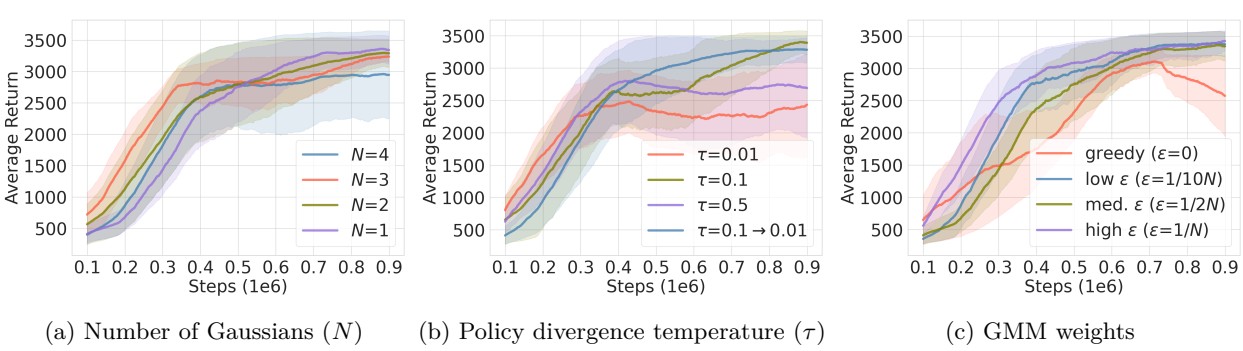

(a) Number of Gaussians ($N$)      (b) Policy divergence temperature ($\tau$)      (c) GMM weights

Figure 12: Ablation curves for the 'Hopper-v3' domain

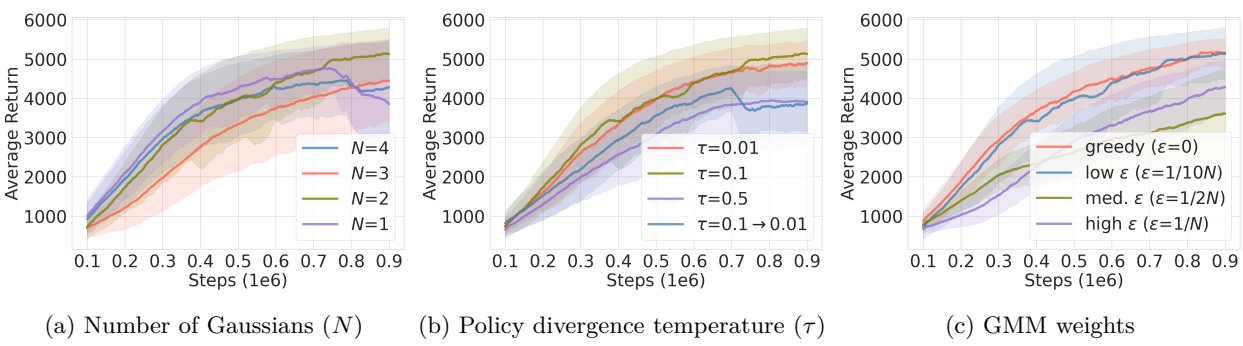

(a) Number of Gaussians ($N$)      (b) Policy divergence temperature ($\tau$)      (c) GMM weights

Figure 13: Ablation curves for the 'Ant-v3' domain

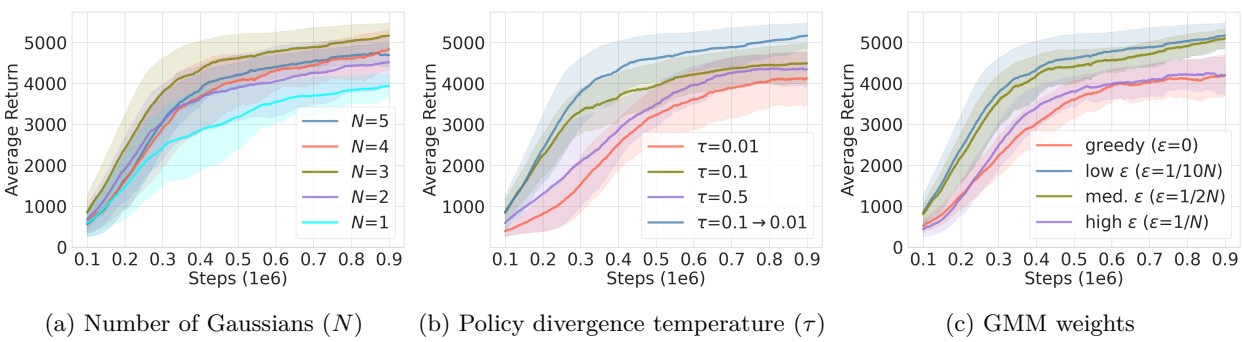

(a) Number of Gaussians ($N$)      (b) Policy divergence temperature ($\tau$)      (c) GMM weights

Figure 14: Ablation curves for the 'Walker2d-v3' domain

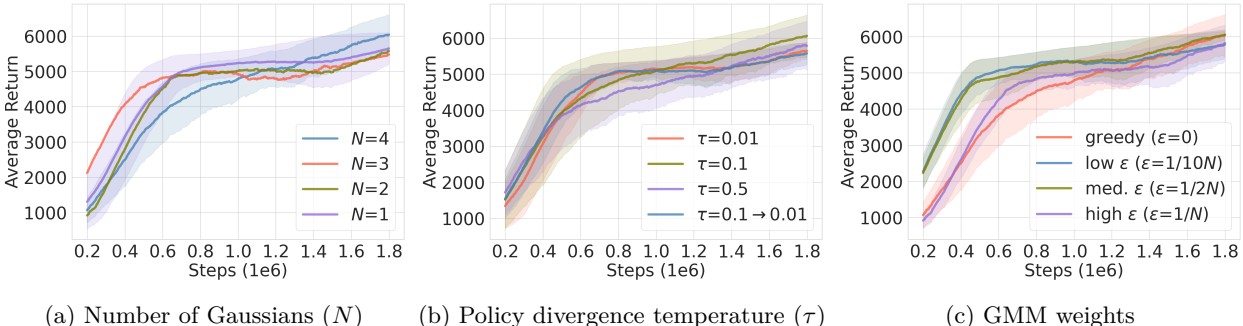

(a) Number of Gaussians ($N$)  (b) Policy divergence temperature ($\tau$)  (c) GMM weights

Figure 15: Ablation curves for the 'Humanoid-v3' domain

