# OpenReview forum: "Comparing Deterministic and Soft Policy Gradients for Optimizing Gaussian Mixture Actors"
_TMLR — Accepted by TMLR_

### Review · Reviewer_SD3W · 2024-10-11

**Summary Of Contributions:**

* The paper proposes Gaussian Mixture Deterministic Policy Gradient (Gamid-PG), which applies deterministic policy gradients to Gaussian Mixture Models (GMMs) in reinforcement learning. This approach contrasts with the commonly used stochastic gradient-based methods for optimizing GMMs.

* The paper conducts an extensive comparison between deterministic and stochastic policy gradients for GMMs across multiple benchmark tasks, demonstrating that deterministic policy gradients can outperform stochastic ones in dense-reward tasks.

**Audience:**

Yes

**Claims And Evidence:**

Yes

**Requested Changes:**

* I approciate the authors by providing figures to show the training procedure on MuJoCo. However, since there are multiple methods to be compared, it would be better if the author also provide some tables to report the numbers of each task. In this way, we can clearly see the performance of the proposed method and baselines.

* As I mentioned in the weakness, I'm wondering if the authors can provide more experiments on more environments.

**Strengths And Weaknesses:**

Strength:
* The paper provides a comprehensive and clear introduction and related works to demonstrate the motivation. The whole paper is well written in general.

* The authors provide comprehensive experimental details and results, enhancing the study's reproducibility and reliability.

Weakness:
* Although the authors claim the method relies on dense and accurate reward environment, the experiments are mainly on relatively simple tasks in MuJoCo. Can author add more experiments on more complex environment, e.g. Antmaze or beyond MuJoCo? Since in real life it is hard to have accurate stepwise reward, it would be great if the method can be proved working on more diverse tasks.

---

> ### Author Response · Authors · 2024-10-28
>
> We thank the reviewer for their insightful comments and suggestions.
> Please see below our proposed revisions and answers.
>
> Requested changes #1:
>
> We have now included a table reporting the mean performance and the 1-standard deviation (+/-) for Gamid and the baseline algorithms for all domains (Page 11, Table 1). The best-performing RL algorithms have been highlighted in bold. Note that per the suggestion from reviewer tB8P, we have included the mean and tolerance values (shaded regions) in the learning curves (Figure 2). By contrast, we opted to present the 1-standard deviation values in the table instead for clarity (as tolerance values may be asymmetric around the mean). The table offers complementary insights to the learning curves where we see that Gamid outperforms the baselines on 5/6 MuJoCo tasks. On the sparse-reward Fetch tasks, stochastic gradient-based methods (SAC and SACM) outperform TD3 and Gamid. However, Gamid outperforms TD3 (single Gaussian actor) in 3/3 Fetch tasks.
>
> Requested changes #2:
>
> Our choice to focus on MoJoCo environments for evaluation follows from previous work that presented benchmark continuous control RL algorithms [1] [2] [3]. Nonetheless, we did validate Gamid on toy continuous control tasks from the OpenAI gym library. Specifically, we validate its performance in "CartPole-v1" (dense reward), "LunarLanderContinuous-v2" (dense reward), and "MountainCarContinuous-v0" (sparse reward). Learning trends in these domains paint a similar picture to those obtained in the MuJoCo tasks. That is, Gamid performs better than, or on par with, the baseline approaches. Due to their simplicity, these environments are not commonly reported in contemporary publications. However, we would be happy to include them in the Appendix section if the reviewer feels strongly about it. Moreover, beyond the MuJuCo environments, please note that we report results on 3 sparse reward tasks from the "Fetch" environments. We used HER [4] on top of the RL algorithms since prior work has shown that it is necessary to learn successful policies in these environments. Following the reviewer's specific domain request (AntMaze), we ran experiments on both the "AntMaze" and "Adroit" sparse-reward environments and observed that neither the baselines nor Gamid presented meaningful learning. These results are in line with prior work showing that, in order to achieve meaningful learning in these environments, existing RL algorithms require extra guidance in the form of intrinsic rewards, demonstrations, or a curriculum [5] [6] [7]. These "extra guidance" enhancements are tangential to the current focus of the paper. Nonetheless, we speculate Gamid, being an off-policy algorithm, can benefit from such enhancements. Such a study is left for future work.
>
> [1] Lillicrap, T. P. "Continuous control with deep reinforcement learning." arXiv preprint arXiv:1509.02971 (2015).
>
> [2] Fujimoto, Scott, Herke Hoof, and David Meger. "Addressing function approximation error in actor-critic methods." International conference on machine learning. PMLR, 2018.
>
> [3] Haarnoja, Tuomas, et al. "Soft actor-critic: Off-policy maximum entropy deep reinforcement learning with a stochastic actor.'' International conference on machine learning. PMLR, 2018.
>
> [4] Andrychowicz, Marcin, et al. "Hindsight experience replay.'' Advances in neural information processing systems 30 (2017).
>
> [5] Uchendu, Ikechukwu, et al. "Jump-start reinforcement learning.'' International Conference on Machine Learning. PMLR, 2023.
>
> [6] Hussenot, Léonard, et al. "Hyperparameter selection for imitation learning." International Conference on Machine Learning. PMLR, 2021.
>
> [7] Huang, Zhiao, et al. "Reparameterized policy learning for multimodal trajectory optimization." International Conference on Machine Learning. PMLR, 2023.

---

### Review · Reviewer_tB8P · 2024-10-20

**Summary Of Contributions:**

The paper presents a method for training a DDPG-based Actor-Critic agent where the policy is parameterized using a Gaussian Mixture Model. Furthermore the agents trained using this algorithm are compared against TD3, which is equivalent to the presented method when the mixture has a single Gaussian, and SAC and its own distributional variants, which are seen as a different family of off-policy Actor-Critic methods which are based on Max Entropy Reinforcement Learning. The authors claim that by adding GMMs to a DDPG formulation they can get benefits from MaxEnt Actor-Critcs and from Deterministic Actor-Critics.

**Audience:**

Yes

**Broader Impact Concerns:**

No concern.

**Claims And Evidence:**

Yes

**Requested Changes:**

# Clarification Changes
+ It would be great if the authors could add a couple of sentences comparing with D4PG, especifically on the merits of using distributional RL on the Critic vs the Policy. There's no need to run experiments for the comparison, especially given the many additional differences between Gamid-PG and D4PG. The comparison is necessary for approval;
+ In section 4.1 the authors state "We attribute this ability to the entropy maximization term in Gamid, which encourages the GMM components to spread out" why wouldn't this also apply to SAC and lead it to also achieve optimal performance? Also adding SACM to this comparison could potentially help shed a light on the difference of policies parameterized by a Gaussian vs a GMM. The explanation would be necessary for approval, the extra comparison is encouraged;
+ In Figures 3g-i PMOE and PPO don't seem to be benchmarked in the Fetch tasks, while they were benchmarked in the Mujoco tasks, is there any reason for that? An explanation is necessary for approval, but additional experiments are not.

# Validity Changes
 + The number of runs is small as mentioned previously, and we don't know whether the performance of agents in the tasks is normally distributed, as such reporting mean and standard deviation (by plotting mean perfomance and using a 2-std shaded region) may be misleading, and so we request the authors to instead plot the mean agent and tolerance intervals as described in [1]. This change is necessary for approaval;
+ For the toy bandit environment in Figure 1 we request the authors tune the hyperparameters for TD3/DDPG and SAC as they were tuned for Gamid-PG, and that the exploratory phase described in Section 7 of [1] be used for SAC and TD3/DDPG, as it seems to be fundamental to get good performance from the algorithms. This change is also necessary for approval.

[1]: Empirical Design in Reinforcement Learning, Patterson et al, https://arxiv.org/abs/2304.01315

**Strengths And Weaknesses:**

# Strenghts
+ The paper is clear in the presentation of the proposed algorithm;
+ The authors introduce well the idea of using distributional policies in MaxEnt Actor-Critics and use that to motivate their approach.

# Weaknesses
+ It's not as clear why would using a distributional parametrization of the policy of a Deterministic Actor-Critic would bring benefits, especially vs using such parametrization for the Critic (as is done in D4PG);
+ While the algorithm introduces new hyperparameters that are tuned the comparisons against other algorithms use just standard hyperparameters with no extra tuning, giving an unfair advantage to Gamid-PG in the comparisons;
+ The number of runs of each algorithm is small (n = 5), which can hamper the validity of the results when just mean and standard deviation are reported.

---

> ### Author Response · Authors · 2024-10-30
>
> We thank the reviewers for their detailed comments and constructive feedback. We clarify the technical concerns below along with our proposed modifications.
>
> **1. Requested changes:** *"It would be great if the authors ... Gamid-PG and D4PG. The comparison is necessary for approval."*
>
> To clarify the advantages of fitting a GMM over the policy (as in Gamid) instead of using distributional RL on the critic (as in D4PG), we have added the following text under Section 2.2: "These approaches, on the other hand, still converge on a deterministic policy and, thus, might fail to capture diverse modes of optimal behavior, i.e., while the critic ($Q$-network) can capture multiple modes of optimality in the $Q$-value space, the (unimodal) actor is limited in its ability to do the same.
>
> Note that a similar explanation has also been provided in the SACM paper [1], albeit for the case of stochastic policy gradients. They state: "Complex control tasks are more likely to elicit non-unimodal value functions. In such situations, unimodal parameterized policies run the risk of converging to suboptimal local minima. To explore both extremes of Q, we need to allow high entropy by forcing high entropy regularization. In contrast, a mixture model readily fits both centers without suffering high entropy. In this case, each component in the mixture can latch to a different extremum of Q. The resultant mixture policy is more diverse, and therefore robust since it assigns proportional weights to the two local maxima of Q."
>
> **2. Requested changes:** *"In section 4.1 the authors ... Gaussian vs a GMM. The explanation would be necessary for approval, the extra comparison is encouraged.*
>
> Following the reviewer's concern, we have now added plots corresponding to SACM in Figure 1. We used a GMM composed of 5 Gaussians with means initialized around 0 and a learnable entropy coefficient with a target entropy of -1.0. We observe that SACM also fails to converge on the optimal mode.
>
> In order to explain why SAC and SACM fail to converge on the global optimum, we have added the following text to the revised version of our paper under Section 4.1: "When examining the performance of SACM (with 5 Gaussians), we notice that it also converges on the suboptimal action, despite training a GMM actor. While SACM uses a GMM-based actor, similar to Gamid, it trains it using stochastic gradients, in contrast to Gamid. These results suggest that when training a GMM actor, deterministic gradients can be more effective in escaping local optimum when compared to stochastic gradients.
>
> The inability of a stochastic gradient approach (SAC, SACM) to capture the optimal mode is identified in prior work [2] from which we adopted the 1-D continuous bandit environment. They provide the following explanation: ``The Gaussian policy, initialized at 0 with a large standard deviation, can cover the whole solution space. However, the gradient with-respect-to $\mu$ is positive, which means the action probability density will be pushed towards the right, as the expected return on the right side is larger than the left side, although the left side contains a higher extreme value. As a result, the policy will move right and get stuck at the local optimum with a low chance of jumping out''. We agree with the intuition provided in this explanation which highlights the limitations of stochastic gradient-based actor optimization in escaping local optimum."
>
> **3. Requested changes:** *"In Figures 3g-i PMOE and PPO ... but additional experiments are not."*
>
> Following the reviewer's concern we added the following text in Section 4.3: "None of the baselines results in meaningful learning in the Fetch tasks due to sparse rewards. Consequently, we follow prior work (Ibarz et al., 2021; Raffin et al., 2021; Bajaj et al., 2022) and combine them with Hindsight Experience Replay (HER) (Andrychowicz et al., 2017). Note that PPO and PMOE are not straightforward to combine with HER and are thus omitted for these tasks.~\footnote{No prior work combined PPO or PMOE with HER to the best of our knowledge.}"
>
> **4. Requested changes:** *"The number of runs is small as ... tolerance intervals as described in [1]. This change is necessary for approval."*
>
> Note that we follow popular RL algorithms [7] [8] [9] that report the mean performance and 2-standard deviation over 5 seeds. Nonetheless, we thank the reviewer for making us aware of prior work [10] that outlines better practices in reporting empirical results in RL. Per the reviewer's suggestion, we have now modified Figure 3 to include the mean agent and the tolerance intervals with $\alpha$ set to 0.95 and $\beta$ set to 0.7 as suggested in [10] instead of the 2-standard deviation.

---

> > ### Comment · Reviewer_tB8P · 2024-10-31
> > **Figure 3 hasn't been changed**
> >
> > We thank the authors for addressing our comments, the paper is now almost in a state that we would recommend acceptance, but looking at the previous version of the paper it looks like you changed figure 3's legend, but not the image itself, after fixing that we'll recommend the paper be accepted.

---

> ### Author Response · Authors · 2024-10-30
>
> **5. Requested changes:** *For the toy bandit environment ... get good performance from the algorithms. This change is also necessary for approval.*
>
> We adapted the 1-D continuous bandit environment introduced in prior work [2]. We used the recommended hyperparameters for setting up SAC as listed in [2]. For the noise in DDPG, we used a Gaussian distribution with a mean of 0 and a fixed standard deviation of 0.1, the recommended setting for MuJoCo tasks in RL Baselines3 Zoo [11]. This toy environment is designed to show a possible shortcoming of RL algorithms using single Gaussian-based policy representations in capturing the optimal mode in a multimodal reward distribution. Using an initial random exploratory phase will be counterproductive in analyzing the effectiveness of the inherent exploratory capabilities of SAC, SACM, and TD3/DDPG in this case. We also emphasize that such an exploratory phase was also omitted in [2]. These details have been now added under Appendix B.
>
> **Weakness:** *While the algorithm introduces new hyperparameters ...  Gamid-PG in the comparisons.*
>
> We would like to clarify that the reported results on SAC, TD3, and PPO use optimized hyperparameters. Specifically, we used
> the open-source implementations provided in Stable Baselines3 [11], a popular and widely accepted RL library. We used the tuned hyperparameter values for SAC, TD3, and PPO that have been provided in the RL Baselines3 Zoo library (\url{https://github.com/DLR-RM/rl-baselines3-zoo/tree/v1.6.2/hyperparams}) which is built on top of Stable Baselines3. For SACM and PMOE we used hyperparameters recommended in [1] and [12] respectively for the MuJoCo tasks. For the Fetch tasks, we tuned the hyperparameters of SACM using Optuna [13]. Consequently we argue that the hyperparameters for all the baselines (SAC, TD3, PPO, SACM, and PMOE) are highly tuned and that no unfair advantage was provided to Gamid-PG. These details are now added under  Appendix B.
>
> [1] Baram, Nir, et al. "Maximum entropy reinforcement learning with mixture policies." arXiv preprint arXiv:2103.10176 (2021).
>
> [2] Huang, Zhiao, et al. "Reparameterized policy learning for multimodal trajectory optimization." International Conference on Machine Learning. PMLR, 2023.
>
> [3] Trott, Alexander, et al. ``Keeping your distance: Solving sparse reward tasks using self-balancing shaped rewards.'' Advances in Neural Information Processing Systems 32 (2019).
>
> [4] Rengarajan, Desik, et al. ``Reinforcement learning with sparse rewards using guidance from offline demonstration.'' arXiv preprint arXiv:2202.04628 (2022).
>
> [5] Uchendu, Ikechukwu, et al. ``Jump-start reinforcement learning.'' International Conference on Machine Learning. PMLR, 2023.
>
> [6] Andrychowicz, Marcin, et al. ``Hindsight experience replay.'' Advances in neural information processing systems 30 (2017).
>
> [7] Lillicrap, T. P. ``Continuous control with deep reinforcement learning.'' arXiv preprint arXiv:1509.02971 (2015).
>
> [8] Haarnoja, Tuomas, et al. ``Soft actor-critic: Off-policy maximum entropy deep reinforcement learning with a stochastic actor.'' International conference on machine learning. PMLR, 2018.
>
> [9] Wang, Ziyu, et al. "Sample efficient actor-critic with experience replay." arXiv preprint arXiv:1611.01224 (2016).
>
> [10] Patterson, Andrew, et al. "Empirical design in reinforcement learning." arXiv preprint arXiv:2304.01315 (2023).
>
> [11] Raffin, Antonin, et al. "Stable-baselines3: Reliable reinforcement learning implementations." Journal of Machine Learning Research 22.268 (2021): 1-8.
>
> [12] Ren, Jie, et al. "Probabilistic mixture-of-experts for efficient deep reinforcement learning." arXiv preprint arXiv:2104.09122 (2021).
>
> [13] Akiba, Takuya, et al. "Optuna: A next-generation hyperparameter optimization framework." Proceedings of the 25th ACM SIGKDD international conference on knowledge discovery \& data mining. 2019.

---

> ### Author Response · Authors · 2024-11-01
> **Figure 4 and Section 4.3 updated**
>
> Thank you for pointing out the issue with now Figure 4 and apologies for our oversight. Our revised version now includes the tolerance intervals in Figure 4 with $\alpha=0.05$ and $\beta=0.7$ as suggested in [1]. Compared to the previous figure (showing standard deviations), the tolerance intervals for Gamid and the baselines are wider. Taking these new observations into account, we have updated Section 4.3 to include the following text: "**Performance consistency.** In terms of the post-training performance, we observe that Gamid consistently performs at least as well as TD3 on both dense-reward (MuJoCo) and sparse-reward (Fetch) tasks. It outperforms TD3 on 8/9 tasks while performing on par with it on ‘Ant-v3’. These results suggest that, while training with deterministic gradients, utilizing a GMM-based actor (as in Gamid) is consistently advantageous with-respect-to returns when compared to utilizing a single-Gaussian actor (as in TD3)."
>
> [1] Patterson, Andrew, et al. "Empirical design in reinforcement learning." arXiv preprint arXiv:2304.01315 (2023).

---

> > ### Comment · Reviewer_tB8P · 2024-11-04
> > **I don't see how you can claim Gamid outperforms TD3 in 8/9 tasks**
> >
> > From the figure I see clear overlaps of the performance curves of TD3 and Gamid in Hopper-v3, Ant-v3, Humanoid-v3, and FetchSlide-v1, shouldn't then the claim be that Gamid outperforms TD3 in 5/9 tasks?

---

> > > ### Author Response · Authors · 2024-11-04
> > > **Average performance and student's t-test**
> > >
> > > We clarify that our claims were based on the post-training average performance. In order to avoid ambiguity, we (1) revised the text (revision provided below) and (2) added a student's t-test analysis.
> > >
> > > "In terms of the average performance, Gamid outperforms TD3 on 8/9 tasks (see Table 1). We also report an independent two-sample t-test (Cressie & Whitford, 1986) with the p-value significance level set to 0.05 comparing Gamid and TD3. The results indicate that the advantage of Gamid over TD3 is statistically significant in 4/9 tasks (‘HalfCheetah-v3’, ‘Walker2d-v3’, ‘FetchPush-v1’, and ‘FetchPickAndPlace-v1’). For the rest of the domains, the performance difference is not statistically significant."

---

> ### Comment · Reviewer_tB8P · 2024-11-04
> **Decision changed to accept**
>
> After the modifications I believe the claims in the paper are warranted and so it meets TLMR criteria for acceptance. I thank the authors for the fruitful discussion and for the modifications to the paper.

---

### Review · Reviewer_byV9 · 2024-10-21

**Summary Of Contributions:**

The paper studies using a Gaussian Mixture Model to represent the actor in an actor-critic setup. Prior work has found mixed results from using a GMM.

This work reexamines using GMMs, within a DDPG variant. Instead of using a single Gaussian, the actor is a mixture of Gaussians. Traditional DDPG uses a deterministic actor and samples exploration noise from a Gaussian centered around the deterministic actor. In this work the sampling is from the learnt GMM. Finally, there is an added term to encourage diversification of the GMM. To me this is the main change compared to prior work, or compared to just using a GMM within SAC.

When computing the Q-target, the next Q-value is taken by maximizing over the $N$ Gaussian parameters. When updating the actor, each Gaussian's mean is updated separately, with an extra diversity term $D_{KL}(i || N \setminus i)$, indicating the KL between the $i$th Gaussian and the mean of every other Gaussian in the mixture. There isn't a closed form for this, but there is a closed form approximation that is straightforward to compute, especially when all Gaussians share the same variance.

For exploration, the action is sampled according to epsilon-greedy: the maximizing Gaussian in the mixture is selected $1-\epsilon$ of the time, and the Gaussian is picked randomly otherwise.

**Audience:**

Yes

**Claims And Evidence:**

Yes

**Requested Changes:**

Would appreciate a bit more analysis on what mixtures have been learned in the MuJoCo experiments. Most of the plots are for expected return, and although that is important, it would be good to see if the diversity KL term is successfully pushing the Gaussians to be diverse.

**Strengths And Weaknesses:**

SAC with a GMM slightly outperforms a baseline SAC method when trained according to this paper, suggesting that the KL divergence term or exploration strategy are the reason prior works did not see much benefit from using a GMM. The DDPG variant, Gamid, has more mixed results, although this is roughly in line with what I've seen in prior RL algorithms papers.

There are an extensive set of ablations inspecting how the number of Gaussians, temperature, etc. affect the method, indicating benefit a small number of Gaussians > 1 and a small but non-zero epsilon greedy exploration.

Overall, although I am worried about the extra hyperparams introduced by this method, I do think it suggests some practical implementation choices for how to more effectively use GMMs within RL.

---

> ### Author Response · Authors · 2024-10-30
>
> We thank the reviewer for their supportive feedback. Please see below our proposed revisions with regard to the requested changes.
>
> Per the reviewer's suggestion, we have now included (1) curves of $\Sigma D_{var}$ at each training step, and (2) visualizations of the means of the GMM Gaussians at various training steps on three representative environments, "Hopper-v3", "HalfCheetah-v3", and "Walker2d-v3" in subsection 4.1.1 and Appendix D.1. We include results for two extreme $\tau$ values (0 and 10) and also provide results for the optimized $\tau$ values that are reported in Table 5. Figure 2 presents the $\Sigma D_{var}$ curves for the different $\tau$ values. We observe that the curve corresponding to $\tau = 10$ is consistently higher than the one corresponding to $\tau = 0$, which is in line with the expected trends.
>
> Addressing the reviewer's concern more generally, we added the following text to the paper under section 4.1.1: "We analyze the diversity between the Gaussians in Gamid captured by the approximated KL-divergence. Specifically, we utilize Equation (4) for approximating the sum of $KL$-divergence over all Gaussians with respect to the GMM. That is, we define $\Sigma D_{var} = \sum_{i=1}^{N} D_{KL}( i \Vert (N\setminus i) )$ (see definition from Line 11 in Algorithm 1). Figure 2 contains plots of $\Sigma D_{var}$ at each training step on three representative MuJoCo tasks (Todorov et al., 2012), ‘Hopper-v3’, ‘HalfCheetah-v3’, and ‘Walker2d-v3’ with increasing order of action dimensionality. We include results for two extreme values of $\tau$ (0 and 10) to highlight their effect on $\Sigma D_{var}$. We expect $\Sigma D_{var}$ to be relatively higher for higher values of $\tau$ since a higher value of $\tau$ maximizes $\Sigma D_{var}$ (Line 11 in Algorithm 1). The $\Sigma D_{var}$ curves for all the tasks in Figure 2 follow the expected trend as the curve corresponding to $\tau$ = 10 is consistently higher than the one corresponding to $\tau$ = 0. For comparison, we also include curves for the tuned $\tau$ values ($\tau$ = opt.) as listed in Table 5 which we observe to be between the curves corresponding to $\tau$ = 0 and $\tau$ = 10. These curves are closer to $\tau$ = 0 since the tuned values are closer to zero at (0.01 – 0.3). We further visualize the evolution of the GMM Gaussian component means in Gamid during different steps of the training stage. See Appendix D.1 for full details and results."

---

### Decision · Action_Editor_PWnB · 2024-11-12

**Recommendation:** Accept as is

**Comment:**

All of the reviewers believe that the the contribution is novel and the claims have been supported, and after a few rounds of discussion, eventually proposed a mix of Accept / Learning Accept.

Specifically, the main changes were:
  * Reviewer tB8P asked for more fair hyperparameter tuning over baselines, and more rigorous plotting. There was also confusion on why distributional RL would help D4PG, but this was cleared by the authors in the paper.
  * Reviewer SD3W asked for experiments on high-exploration environments (e.g. Ant-Maze).  Authors responded that GMM's don't completely solve the exploration problem.

Judging by the potential impact however, this paper currently does not warrant a featured certification, as the consensus has been lukewarm.

**Audience:**

Yes, the paper is relevant for anybody working in standard reinforcement learning settings.

**Claims And Evidence:**

Yes. The authors propose using Gaussian Mixture action distributions over Deterministic Policy Gradient (DPG), and show that an additional KL exploration term was necessary to avoid mean collapse (encountered in previous works).

Authors establish the benefits of this approach on control tasks over regular DPG methods, while also ablating some underperforming cases (e.g. on sparse reward tasks) against stochastic gradient approaches.